# Addressing Long-Horizon Tasks by Integrating Program Synthesis and State Machines

## Abstract

Deep reinforcement learning excels in various domains but lacks generalizability and interoperability. Programmatic RL (Trivedi et al., 2021; Liu et al., 2023) methods reformulate solving RL tasks as synthesizing interpretable programs that can be executed in the environments. Despite encouraging results, these methods are limited to short-horizon tasks. On the other hand, representing RL policies using state machines (Inala et al., 2020) can inductively generalize to long-horizon tasks; however, it struggles to scale up to acquire diverse and complex behaviors. This work proposes Program Machine Policies (POMPs), which bridge the advantages of programmatic RL and state machine policies, allowing for representing complex behaviors and addressing long-horizon tasks. Specifically, we introduce a method that can retrieve a set of effective, diverse, compatible programs. Then, we use these programs as modes of a state machine and learn a transition function to transition among mode programs, allowing for capturing long-horizon repetitive behaviors. Our proposed framework outperforms programmatic RL and deep RL baselines on various tasks and demonstrates the ability to inductively generalize to even longer horizons without any fine-tuning. Ablation studies justify the effectiveness of our proposed search algorithm for retrieving a set of programs as modes.

## 1 Introduction

Deep reinforcement learning (deep RL) has recently achieved tremendous success in various domains, such as controlling robots (Gu et al., 2017; Ibarz et al., 2021), playing strategy board games (Silver et al., 2016; 2017), and mastering video games (Vinyals et al., 2019; Wurman et al., 2022). However, the black-box neural network policies learned by deep RL methods are not human-interpretable, posing challenges in scrutinizing model decisions and establishing user trust (Lipton, 2016; Shen, 2020). Moreover, deep RL policies often suffer from overfitting and struggle at generalizing to novel scenarios (Zhang et al., 2018; Cobbe et al., 2019), limiting their applicability in the context of most real-world applications.

To address these issues, Trivedi et al. (2021); Liu et al. (2023) explored representing policies as programs, which detail task-solving procedures in a formal programming language and can be executed to solve tasks described by Markov Decision Processes (MDPs). Such *program policies* are human-readable and demonstrate significantly improved generalizability that allows for learning in a smaller state space and zero-shot generalizing to larger state spaces. Despite the encouraging results, these methods are limited to synthesizing concise programs (*i.e.*, shorter than 120 tokens) which can only tackle short-horizon tasks (*i.e.*, less than 400 time steps) (Liu et al., 2023).

To solve tasks requiring inductively generalizing to longer horizons, Inala et al. (2020) proposed representing a policy using a state machine. By learning to transition between a set of modes encapsulating actions corresponding to specific states, such *state machine policies* can naturally model long-horizon repetitive behaviors. Yet, this approach is constrained by highly simplified, task-dependent grammar that can only structure constant or proportional primitive actions. Additionally, its teacher-student training scheme requires model-based trajectory optimization with an accurate environment model, which can often be challenging to attain in practice (Polydoros & Nalpantidis, 2017).

This work aims to bridge the best of both worlds — the interpretability and scalability of *program policies* and the inductive generalizability of *state machine policies*. We propose **pro**gram **m**achine **p**olices (POMP), which learns a state machine upon a set of diverse programs. Intuitively, POMP

can tackle long-horizon tasks by repetitively transitioning between programs and is scalable since each mode is a high-level skill described by a program instead of the agent's primitive action.

We propose a three-stage framework to learn such a program machine policy. (1) **Constructing a program embedding space**: To establish a program embedding space that smoothly, continuously parameterizes programs with diverse behaviors, we adopt the method proposed by Trivedi et al. (2021). (2) **Retrieving a diverse set of effective and reusable programs**: Then, we introduce a searching algorithm which can retrieve a set of programs from the learned program embedding space. Each program can be executed in the MDP and achieve satisfactory performance; more importantly, these programs are compatible and can be sequentially executed in any order. (3) **Learning the transition function**: To alter between a set of programs as state machine modes, the transition function takes the current environment state and the current mode (*i.e.*, program) as input, and predicts the next mode to transition to. We propose to learn this transition function using RL via maximizing the task rewards from the MDP.

To evaluate our proposed framework POMP, we adopt the Karel domain (Pattis, 1981), which characterizes an agent that navigates a grid world and interacts with objects. POMP outperforms programmatic reinforcement learning and deep RL baselines on existing benchmarks proposed by Trivedi et al. (2021); Liu et al. (2023). We design a new set of tasks that features long-horizon tasks on which POMP demonstrates superior performance and the ability to inductively generalize to even longer horizons without any fine-tuning. Ablation studies justify the effectiveness of our proposed search algorithm for retrieving a set of programs as modes.

## 2 RELATED WORK

**Programmatic Reinforcement Learning.** Programmatic reinforcement learning methods (Choi & Langley, 2005; Winner & Veloso, 2003; Liu et al., 2023) explore structured representations for representing RL policies, including decision trees (Bastani et al., 2018), state machines (Inala et al., 2020), symbolic expressions (Landajuela et al., 2021), and programs (Verma et al., 2018; 2019; Aleixo & Lelis, 2023). Trivedi et al. (2021); Liu et al. (2023) attempted to produce policies described by domain-specific language programs to solve simple RL tasks. We aim to take a step toward addressing complex, long-horizon, repetitive tasks.

**State Machines for Reinforcement Learning.** Recent works adopt state machines to model rewards (Icarte et al., 2018; Toro Icarte et al., 2019; Furelos-Blanco et al., 2023; Xu et al., 2020), segment tasks (Hasanbeig et al., 2021; Furelos-Blanco et al., 2021), or achieve inductive generalization (Inala et al., 2020). Prior works explore using symbolic programs (Inala et al., 2020) or neural networks (Icarte et al., 2018; Toro Icarte et al., 2019; Hasanbeig et al., 2021) as modes (*i.e.*, states) in state machines. However, the learned policy of such designs is not easy to interpret compared to DSL programs. On the contrary, our goal is to exploit human-readable programs for each mode of state machines so that the resulting state machine policies are more easily interpreted.

## 3 PROBLEM FORMULATION

Our goal is to devise a framework that can produce a program machine policy (POMP), a state machine whose modes are programs structured in a domain-specific language, to address complex, long-horizon tasks described by Markov Decision Processes (MDPs). To this end, we first synthesize a set of task-solving, diverse, compatible programs as modes, and then learn a transition function to alter between modes.

**Domain Specific Language.** This work adopts the domain-specific language (DSL) of the Karel domain Bunel et al. (2018); Chen et al. (2019); Trivedi et al. (2021), as illustrated in Figure 1. This DSL describes the control flows as well as the perception and actions of the Karel agent. Actions including `move`, `turnRight`, and `putMarker` define how the agent can interact with the environment. Perceptions, such as `frontIsClear` and `markerPresent`, formulate how the agent observes the environment. Control flows, *e.g.*, `if`, `else`, `while`, enable representing divergent and repetitive behaviors. Furthermore, Boolean and logical operators like `and`, `or`, and `not` allow for composing more intricate conditions. This work uses programs structured in this DSL to construct the modes of a program machine policy.

**Markov Decision Process (MDP).** The tasks considered in this work can be formulated as finite-horizon discounted Markov Decision Processes (MDPs). The performance of a program machine policy is evaluated based on the execution traces of a series of programs selected by the state machine

transition function. The rollout of a program $\rho$ consists of a $T$-step sequence of state-action pairs $\{(s_t, a_t)\}_{t=1, ..., T}$ obtained from a program executor $\mathrm{EXEC}(\cdot)$ that executes program $\rho$ to interact with an environment, resulting in the discounted return $\sum_{t=0}^{T} \gamma^t(r_t)$, where $r_t = \mathcal{R}(s_t, a_t)$ denotes the reward function. We aim to maximize the total rewards by executing a series of programs following the state machine transition function.

**Program Machine Policy (POMP).** This work proposes a novel RL policy representation, a program machine policy, which consists a finite set of *modes* $M = \{m_k\}_{k=1, ..., |M|}$ as internal states of the state machine and a state machine *transition function* $f$ that determines how to transition among these modes. Each *mode* $m_i$ encapsulates a human-readable program $\rho_{m_i}$ that will be executed when this mode is selected during policy execution. On the other hand, the *transition function* $f(m_i, m_j, s)$ outputs the probability of transitioning from the current mode $m_i$ to mode $m_j$ given the current MDP state $s$. To rollout a POMP, the state ma-

```
Program ρ ≔ DEF run m( s m)
Repetition n ≔ Number of repetitions
Perception h ≔ frontIsClear | leftIsClear | rightIsClear |
              markerPresent | noMarkerPresent
Condition b ≔ perception h | not perception h
Action a ≔ move | turnLeft | turnRight |
          putMarker | pickMarker
Statement s ≔ while c( b c) w( s w) | s₁; s₂ | a |
              repeat R=n r( s r) | if c( b c) i( s i) |
              ifelse c( b c) i( s₁ i) else e( s₂ e)
```

Figure 1: **Karel domain-specific language (DSL)**, designed for describing the Karel agent's behaviors.

chine starts at initial mode $m_{\mathrm{init}}$, which will not execute any action, and transits to the next mode $m_{i+1}$ based on the current mode $m_i$ and MDP state $s$. If $m_{i+1}$ equals the termination mode $m_{\mathrm{term}}$, the program machine policy will terminate and finish the rollout. Otherwise, the mode program $\rho_{m_{i+1}}$ will be executed and generates state-action pairs $\{(s_t^i, a_t^i)\}_{t=1, ..., T^i}$ before the state machine transits to the next state $m_{i+2}$.

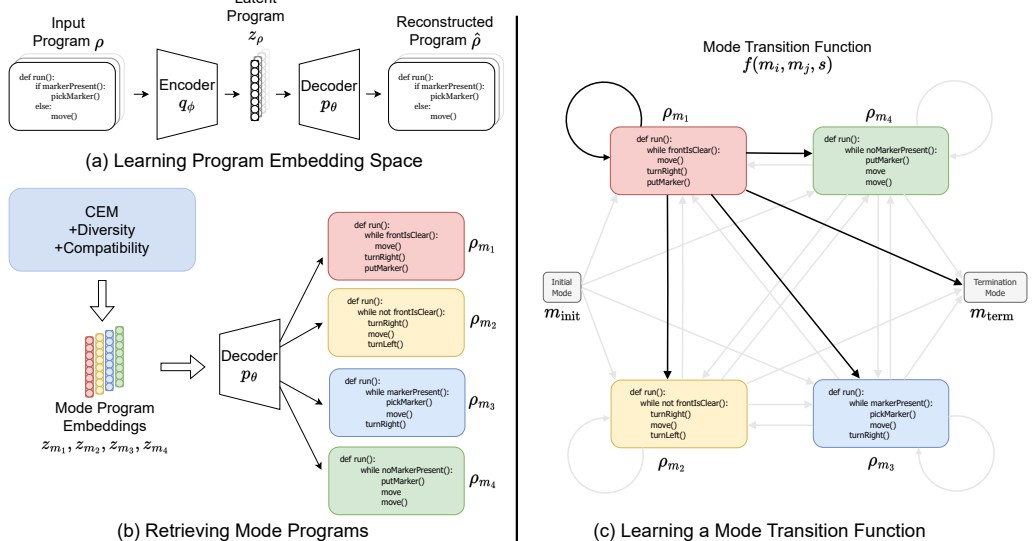

Figure 2: **Learning Program Machine Policy. (a): Learning a program embedding space.** To learn a program embedding space that encodes program semantics behaviors, we follow the framework proposed in Trivedi et al. (2021), employing an encoder $q_\phi$ and a decoder $p_\theta$. **(b): Retrieving mode programs.** After learning the program embedding space, we propose an advanced search scheme built upon the Cross Entropy Method (CEM) to search program embeddings $z_{m1}, z_{m2}, z_{m3}, z_{m4}$ of different skills. These embeddings can be decoded by $p_\theta$ to get the executable programs $\rho_{m_1}, \rho_{m_2}, \rho_{m_3}, \rho_{m_4}$ as policies for the modes of the state machine. **(c): Learning a mode transition function.** Given the current environment state $s$ and the current mode ($m_1$ in this example), the mode transition function predicts the transition probability over each mode of the state machine with the aim of maximizing the total accumulative reward from the environment.

## 4 APPROACH

We design a three-stage framework to produce a program machine policy that can be executed and maximize the return given a task described by an MDP. First, constructing a program embed-

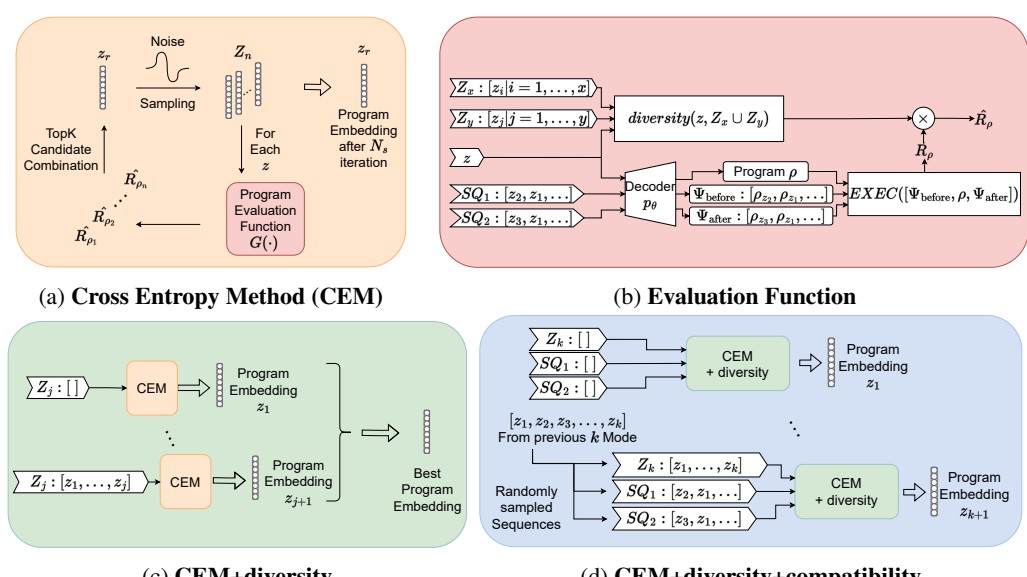

Figure 3: Searching for a set of effective and reusable program embedding that can be decoded to programs of various skills. (a): Using the Cross Entropy Method to search for a program with high execution reward in the learned program embedding space. (b): The program evaluation function $G$ used during program search. (c): Conducting CEM multiple times and selecting the best program. The *diversity multiplier* is used to avoid identical results and increase diversity among multiple CEM searches. (d): Searching for a diverse set of program embeddings that are compatible with each other in terms of execution order $SQ_1$ and $SQ_2$.

ding space that smoothly, continuously parameterizes programs with diverse behaviors is introduced in Section 4.1. Then, Section 4.2 presents a method that retrieves a set of effective, diverse, compatible programs as POMP modes. Given retrieved modes, Section 4.3 describes learning the transition function determining transition probability among the modes. An overview of the proposed framework is illustrated in Figure 2.

## 4.1 CONSTRUCTING PROGRAM EMBEDDING SPACE

We follow the approach and the program dataset presented in Trivedi et al. (2021) to learn a program embedding space that smoothly, continuously parameterizes programs with diverse behaviors. The training objectives include a VAE loss and two losses that encourage learning a behaviorally smooth program embedding space. Once trained, we can use the learned decoder $p_\theta$ to map any program embedding $z$ to a program $\rho_z = p_\theta(z)$ consisting of a sequence of program tokens.

## 4.2 RETRIEVING MODE PROGRAMS

With a program embedding space, we aim to retrieve a set of programs as modes of a program machine policy given a task. This set of programs should satisfy the following properties.

- **Effective**: Each program can solve the task to some extent (*i.e.*, obtain some task rewards).
- **Diverse**: The more behaviorally diverse the programs are, the richer behaviors can be captured by the program machine policy.
- **Compatible**: Sequentially executing some programs with specific orders can potentially lead to improved task performance.

In the following, we present techniques that allow retrieving such an effective, diverse, and compatible set of programs from a learned program embedding space.

### 4.2.1 RETRIEVING EFFECTIVE PROGRAMS

To obtain a task-solving program, we can apply the Cross-Entropy Method (CEM; Schmidhuber et al., 1997), iteratively searching in a learned program embedding space (Trivedi et al., 2021), described below and illustrated in Figure 3a.

(1) Randomly initialize a program embedding vector $z_r$ as the search center.

(2) Add random noises to $z_r$ to generate a population of program embeddings $Z = \{z_i\}_{i=1,\dots,n}$, where $n$ denotes the population size.

(3) Evaluate every program embedding $z \in Z$ with the evaluation function $G$ to get a list of fitness score $[G(z_i)]_{i=1,\dots,n}$.

(4) Average the top k program embeddings in $Z$ according to fitness scores $[G(z_i)]_{i=1,\dots,n}$ and assign it to the search center $z_r$.

(5) Repeat (2) to (4) until the fitness score $G(z_r)$ of $z_r$ converges or the maximum number of steps is reached.

Since we aim to retrieve a set of effective programs, we can define the evaluation function as the program execution return of a decoded program embedding, *i.e.*, $G(z) = \sum_{t=0}^{T} \gamma^t \mathbb{E}_{(s_t,a_t)\sim\text{EXEC}(\rho_z)}[r_t]$. To retrieve a set of $|M|$ programs as the modes of a program machine policy, we can run this CEM search $N$ times, take $|M|$ best program embeddings, and obtain the decoded program set $\{\rho_{z_{r_i}} = p_\theta(z_{r_i})\}_{i=1,\dots,|M|}$. Section B.1 presents more details and the CEM search pseudocode.

### 4.2.2 RETRIEVING EFFECTIVE, DIVERSE PROGRAMS

We will use the set of retrieved programs as the modes of a program machine policy. Hence, a program set with diverse behaviors can lead to a program machine policy representing complex, rich behavior. However, the program set obtained by running the CEM search for $|M|$ times can have low diversity, preventing the policy from solving tasks requiring various skills.

To address this issue, we propose considering previous search results to encourage diversity among the retrieved programs by employing a *diversity multiplier* in the evaluation function $G$. Specifically, during the $(j + 1)$st CEM search, each program embedding $z$ is evaluated by $G(z, Z_j) = (\sum_{t=0}^{T} \gamma^t \mathbb{E}_{(s_t,a_t)\sim\text{EXEC}(\rho_z)}[r_t]) \cdot diversity(z, Z_j)$, where $diversity(z, Z_j)$ is the proposed *diversity multiplier* defined as $Sigmoid(-\max_{z_i \in (Z_j)} \frac{z \cdot z_i}{\|z\|\|z_i\|})$. Thus, the program execution return is scaled down by $diversity(z, Z_j)$ based on the maximum cosine similarity between $z$ and the retrieved program embeddings $Z_j = \{z_i\}_{i=1,\dots,j}$ from the previous $j$ CEM searches. This diversity multiplier encourages searching for program embeddings different from previously retrieved programs. An illustration of this process is shown in Figure 3c.

To retrieve a set of $|M|$ programs as the modes of a program machine policy, we can run this **CEM+diversity** search $N$ times, take $|M|$ best program embeddings, and obtain the decoded program set. The procedure and the search trajectory visualization can be found in Section B.2.

### 4.2.3 RETRIEVING EFFECTIVE, DIVERSE, COMPATIBLE PROGRAMS

Our program machine policy executes a sequence of programs by learning a transition function to select from mode programs. Therefore, these programs need to be compatible with each other, *i.e.*, executing a program following the execution of other programs can improve task performance. Yet, CEM+diversity discussed in Section 4.2.2 searches every program independently.

In order to account for the compatibility among programs during the search, we propose a method, CEM+diversity+compatibility, as illustrated in Figure 3d. When searching the program for the $(k + 1)$st mode, we randomly sample two lists of programs from the determined mode set with $k$ programs: $\Psi_{\text{before}}$ and $\Psi_{\text{after}}$. Then, during each iteration of the CEM search, we compute the return of each program embedding $z$ by sequentially executing this list of programs: $[\Psi_{\text{before}}, \rho_z, \Psi_{\text{after}}]$, where $\rho_z$ denotes the program decoded from $z$. That said, a program achieving a good return is compatible with the $k$ previously found modes under some execution orders.

Note that to compute the diversity multiplier, when running the $(j + 1)$st search with $k$ determined modes, we consider all the $j + k$ previously found program embeddings $Z_j \cup Z_k$ by calculating $diversity(z, Z_j \cup Z_k)$. As a result, the evaluation function is $G(z, Z_j \cup Z_k, \Psi_{\text{before}}, \Psi_{\text{after}}) = R_\rho \cdot diversity(z, Z_j \cup Z_k)$, where $R_\rho$ is the total reward obtained from sequentially executing decoded programs of $\Psi_{\text{before}}$, $z$, and $\Psi_{\text{after}}$ and can be written as follows:

$$
\begin{aligned}
R_\rho = \sum_{i=1}^{|\Psi_{\text{before}}|} \gamma^{i-1} \sum_{t=0}^{T^i} \gamma^t \mathbb{E}_{(s_t,a_t)\sim\text{EXEC}(\Psi_{\text{before}}[i])}[r_t] + \gamma^{|\Psi_{\text{before}}|} \sum_{t=0}^{T} \gamma^t \mathbb{E}_{(s_t,a_t)\sim\text{EXEC}(\rho_z)}[r_t] \\
+ \gamma^{|\Psi_{\text{before}}|+1} \sum_{i=1}^{|\Psi_{\text{after}}|} \gamma^{i-1} \sum_{t=0}^{T^i} \gamma^t \mathbb{E}_{(s_t,a_t)\sim\text{EXEC}(\Psi_{\text{after}}[i])}[r_t].
\end{aligned} \tag{1}
$$

We can run this search for $|M|$ times to obtain a set of programs that are effective, diverse, and compatible with each other, which can be used as mode programs for a program machine policy. More details and the whole search procedure can be found in Section B.3.

### 4.3 LEARNING TRANSITION FUNCTION

Given a set of modes (*i.e.*, programs) $M = \{m_k\}_{k=1, ..., |M|}$, we formulate learning a transition function $f$ that determines how to transition between modes as a reinforcement learning problem aiming to maximize the task return. In practice, we define an initial mode $m_{\text{init}}$ that initializes the program machine policy at the beginning of each episode; also, we define a termination mode $m_{\text{term}}$, which terminate the episode if chosen. Specifically, the transition function $f(m_i, m_j, s)$ outputs the probability of transitioning from the current mode $m_i$ to mode $m_j$, given the current environment $s$.

At $i$-th transition function step, given the current state $s$ and the current mode $m_{\text{current}}$, the transition function predicts the probability of transition to $m_{\text{term}} \cup \{m_k\}_{k=1, ..., |M|}$. We sam-

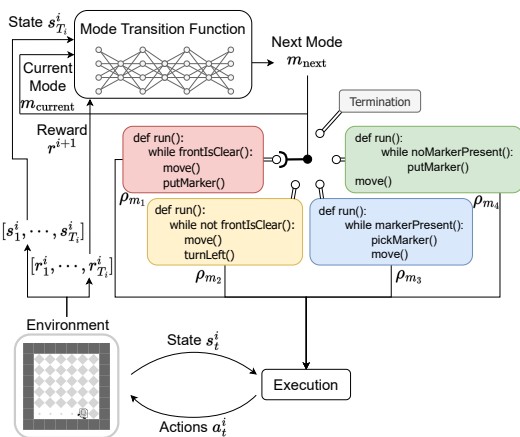

Figure 4: **Program Machine Policy Execution.**

ple a mode $m_{\text{next}}$ based on the predicted probability distribution. If the sampled mode is the termination mode, the episode terminates; otherwise, we execute the corresponding program $\rho$, yielding the next state (*i.e.*, the last state $s^i_{T^i}$ of the state sequence returned by $\text{EXEC}(\rho)$, where $T^i$ denotes the horizon of the $i$-th program execution) and the cumulative reward $r^{i+1} = \sum_{t=1}^{T^i} r^i_t$. Note that the program execution $\text{EXEC}(\rho)$ will terminate after full execution or the number of actions emitted during $\text{EXEC}(\rho)$ reaches 200. We assign the next state to the current state and the next mode to the current mode. Then, we start the next $(i + 1)$st transition function step. This process stops when the termination mode is sampled or a maximum step is reached. Figure 4 illustrates this procedure.

Intuitively, the transition function learns to maximize the total rewards obtained from the entire program machine policy execution $\sum_{t=1}^{H} r^i$, where $H$ denotes the horizon of the execution. To make the transition function interpretable, we adopt approaches proposed in Koul et al. (2019) to extract the state machine structure. Examples of extracted state machines are shown in Section E.

## 5 EXPERIMENTS

We aim to answer the following questions with the experiments and ablation studies. (1) Can our proposed *diversity multiplier* introduced in Section 4.2.2 enhance CEM and yield programs with improved performance? (2) Can our proposed CEM+diversity+compatibility introduced in Section 4.2.3 retrieve a set of programs that are diverse yet compatible with each other? (3) Can the whole proposed framework produce a program machine policy that can be executed and maximize the return given a task described by an MDP and outperform existing methods?

### 5.1 KAREL PROBLEM SETS

To this end, we consider the Karel domain (Pattis, 1981), which is widely adopted in program synthesis (Bunel et al., 2018; Shin et al., 2018; Sun et al., 2018; Chen et al., 2019) and programmatic reinforcement learning (Trivedi et al., 2021; Liu et al., 2023). Specifically, we utilize the KAREL problem set (Trivedi et al., 2021) and the KAREL-HARD problem set (Liu et al., 2023).

The KAREL problem set includes six basic tasks, each can be solved by a short program (less than 45 tokens), with a horizon shorter than 200 steps per episode. On the other hand, the four tasks introduced in the KAREL-HARD problem require longer, more complex programs (*i.e.*, 45 to 120 tokens) with longer execution horizons (*i.e.*, up to 500 actions). Solving the tasks in these two problem sets requires conducting the same behavior repetitively, traversing through multiple rooms, performing some specific actions in particular locations, etc. More details about the two problem sets can be found in Section G and Section H.

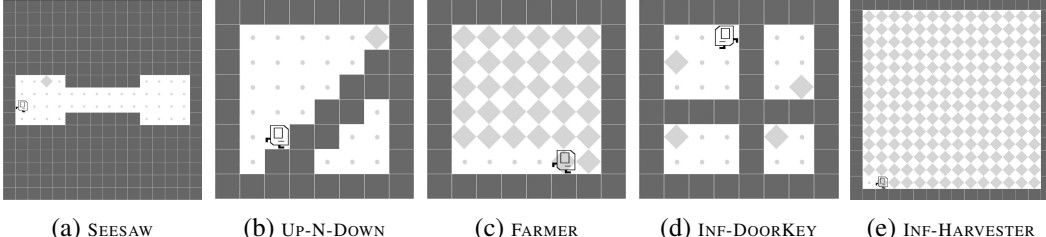

(a) SEESAW     (b) UP-N-DOWN     (c) FARMER     (d) INF-DOORKEY     (e) INF-HARVESTER

Figure 5: **KAREL-LONG Problem Set**: This work introduces a new set of tasks in the Karel domain. These tasks necessitate learning diverse, repetitive, and task-specific skills. For example, in our designed INF-HARVESTER the agent needs to traverse the whole map and pick nearly 400 markers to solve the tasks since the environment randomly generates markers; in contrast, the HARVESTER from the KAREL problem set (Trivedi et al., 2021) can be solved by picking 36 markers.

Table 1: **Comparing CEM and CEM+diversity.** The accumulated rewards of programs retrieved by CEM and CEM+diversity. The mean and standard deviation are evaluated over five random seeds. CEM+diversity outperforms CEM with significantly smaller standard deviations across 8 out of 10 tasks, highlighting the effectiveness and stability of CEM+diversity.

| Method | FOUR CORNER | TOP OFF | CLEAN HOUSE | STAIR CLIMBER | HARVESTER | MAZE | DOOR KEY | ONE STROKE | SEEDER | SNAKE |
|---|---|---|---|---|---|---|---|---|---|---|
| CEM | $0.45 \pm 0.40$ | $0.81 \pm 0.07$ | $0.18 \pm 0.14$ | $\mathbf{1.00} \pm 0.00$ | $0.45 \pm 0.28$ | $\mathbf{1.00} \pm 0.00$ | $\mathbf{0.50} \pm 0.00$ | $\mathbf{0.65} \pm 0.19$ | $0.51 \pm 0.21$ | $0.21 \pm 0.15$ |
| CEM+diversity | $\mathbf{1.00} \pm 0.00$ | $\mathbf{1.00} \pm 0.00$ | $\mathbf{0.37} \pm 0.06$ | $\mathbf{1.00} \pm 0.00$ | $\mathbf{0.80} \pm 0.07$ | $\mathbf{1.00} \pm 0.00$ | $\mathbf{0.50} \pm 0.00$ | $0.62 \pm 0.01$ | $\mathbf{0.69} \pm 0.07$ | $\mathbf{0.36} \pm 0.02$ |

**KAREL-LONG Problem Set.** To evaluate the performance of our proposed framework and existing methods on tasks with extended horizons, we design a new set of Karel tasks, dubbed KAREL-LONG problem set, whose tasks require thousands of steps to be completed. As illustrated in Figure 5, the tasks requires the agent to fulfill extra constraints (*e.g.*, not placing multiple markers on the same spot in FARMER, receiving penalties imposed for not moving along stairs in UP-N-DOWN) and conduct extended exploration (*e.g.*, repetitively locating and collecting markers in SEESAW, INF-DOORKEY, and INF-HARVESTER). More details about the KAREL-LONG tasks can be found in Section I.

### 5.2 CROSS-ENTROPY METHOD WITH DIVERSITY MULTIPLIER

We aim to investigate whether our proposed *diversity multiplier* can enhance CEM and yield programs with improved performance. To this end, for each KAREL or KAREL-HARD task, we use CEM and CEM+diversity to find 10 programs. Then, for each task, we evaluate all the programs and report the best performance in Table 1. The results suggest that our proposed CEM+diversity achieves better performance on most of the tasks, highlighting the improved search quality induced by covering wider regions in the search space with the *diversity multiplier*. Visualized search trajectories of CEM+diversity can be found in Section B.2.

### 5.3 ABLATION STUDY

We propose CEM+diversity+compatibility to retrieve a set of effective, diverse, compatible programs as modes of our program machine policy. This section compares a variety of implementations that consider the diversity and the compatibility of programs when retrieving them.

- **CEM** $\times |M|$: Conduct the CEM search described in Section 4.2.1 $|M| = 5$ times and take the resulting $|M|$ programs as the set of mode programs for each task.

- **CEM+diversity top** $k$, $k = |M|$: Conduct the CEM search with the *diversity multiplier* described in Section 4.2.2 $N = 10$ times and take the top $|M| = 5$ results as the set of mode program embeddings for each task.

- **CEM+diversity** $\times |M|$: Conduct the CEM search with the *diversity multiplier* described in Section 4.2.2 $N = 10$ times and take the best program as the $i^{th}$ mode. Repeat this process $|M| = 5$ times and take all $|M|$ programs as the mode program set for each task.

- **POMP (Ours).**: Conduct CEM+diversity+compatibility (*i.e.*, CEM with the *diversity multiplier* and $R_\rho$ as described in Section 4.2.3) for $N = 10$ times and take the best result as the $i^{th}$ mode. Repeat the above process $|M| = 5$ times and take all $|M|$ results as the set

Table 2: **KAREL-LONG Performance.** Mean return and standard deviation of all methods across the KAREL-LONG problem set, evaluated over five random seeds. By learning a program machine policy with a set of effective, diverse, and compatible mode programs, our proposed framework achieves the best mean reward in SEESAW, UP-N-DOWN, FARMER and performs competitively in INF-DOORKEY and INF-HARVESTER.

| Method | SEESAW | UP-N-DOWN | FARMER | INF-DOORKEY | INF-HARVESTER |
|---|---|---|---|---|---|
| CEM $\times |M|$ | $0.31 \pm 0.18$ | $0.88 \pm 0.12$ | $0.22 \pm 0.00$ | $0.39 \pm 0.46$ | $0.56 \pm 0.12$ |
| CEM+diversity top $k$, $k = |M|$ | $0.09 \pm 0.11$ | $0.72 \pm 0.36$ | $0.23 \pm 0.00$ | $0.92 \pm 0.01$ | $0.71 \pm 0.02$ |
| CEM+diversity $\times |M|$ | $0.47 \pm 0.39$ | $0.76 \pm 0.31$ | $0.24 \pm 0.03$ | $0.89 \pm 0.02$ | $0.66 \pm 0.07$ |
| DRL | $0.00 \pm 0.00$ | $0.00 \pm 0.00$ | $0.55 \pm 0.16$ | $\mathbf{0.97} \pm 0.01$ | $0.45 \pm 0.03$ |
| Random Transition | $0.08 \pm 0.03$ | $0.03 \pm 0.17$ | $0.09 \pm 0.03$ | $0.13 \pm 0.08$ | $0.14 \pm 0.07$ |
| PSMP | $0.25 \pm 0.39$ | $0.00 \pm 0.00$ | $0.58 \pm 0.06$ | $\mathbf{0.97} \pm 0.01$ | $0.59 \pm 0.06$ |
| MMN | $0.10 \pm 0.00$ | $0.00 \pm 0.00$ | $0.57 \pm 0.12$ | $0.87 \pm 0.05$ | $0.41 \pm 0.10$ |
| LEAPS | $0.10 \pm 0.10$ | $0.38 \pm 0.05$ | $0.15 \pm 0.06$ | $0.10 \pm 0.04$ | $0.06 \pm 0.06$ |
| HPRL | $0.00 \pm 0.00$ | $0.00 \pm 0.00$ | $0.40 \pm 0.18$ | $0.15 \pm 0.21$ | $\mathbf{0.85} \pm 0.06$ |
| POMP (Ours) | $\mathbf{0.90} \pm 0.02$ | $\mathbf{0.97} \pm 0.00$ | $\mathbf{0.88} \pm 0.01$ | $0.91 \pm 0.01$ | $0.67 \pm 0.03$ |

of mode program embeddings for each task. Note that the whole procedure of retrieving programs using CEM+diversity+compatibility and learning a program machine policy with retrieved mode programs is essentially our proposed framework, POMP.

We evaluate the quality of retrieved program sets according to the performance of program machine policies learned given these program sets on the KAREL-LONG tasks. The results presented in Table 2 show that our proposed framework POMP outperforms its variants that ignore compatibility among modes on all the tasks. This justifies our proposed CEM+diversity+compatibility for retrieving a set of effective, diverse, compatible programs as modes of our program machine policy.

## 5.4 COMPARING WITH DEEP RL AND PROGRAMMATIC RL METHODS

In this section, we compare our proposed framework and its variant to state-of-the-art deep RL and programmatic RL methods on the KAREL-LONG tasks.

- **Random Transition** uses the same set of mode programs as POMP but with a random transition function (*i.e.*, uniformly randomly select the next mode at each step). The performance of this method examines the necessity to learn a transition function.

- **Programmatic State Machine Policy (PSMP)** learns a transition function as POMP while using primitive actions (*e.g.*, move, pickMarker) as modes. Comparing POMP with this method highlights the effect of retrieving programs with higher-level behaviors as modes.

- **DRL** represents a policy as a neural network and is learned using PPO (Schulman et al., 2017). The policy takes raw states (*i.e.*, Karel grids) as input and predicts the probability distribution over the set of primitive actions, *e.g.*, move, pickMarker.

- **Moore Machine Network (MMN)** Koul et al. (2019) represents a recurrent policy with quantized memory and observations, which can be further extracted as a finite state machine. The policy takes raw states (*i.e.*, Karel grids) as input and predicts the probability distribution over the set of primitive actions, *e.g.*, move, pickMarker.

- **Learning Embeddings for Latent Program Synthesis (LEAPS)** Trivedi et al. (2021) searches for a single task-solving program using the vanilla CEM in a learned program embedding space.

- **Hierarchical Programmatic Reinforcement Learning (HPRL)** Liu et al. (2023) learns a meta-policy, whose action space is a learned program embedding space, to compose a series of programs to produce a program policy.

POMP excels in three of the five tasks we devised, with particular prowess in FARMER and SEESAW while performing competitively in the other two tasks. FARMER requires two distinct skills (*e.g.*, pick and put markers) and the capability to persistently execute one skill for an extended period before transitioning to another. POMP adeptly addresses this challenge due to the consideration of diversity when seeking mode programs, which ensures the acquisition of both skills concurrently.

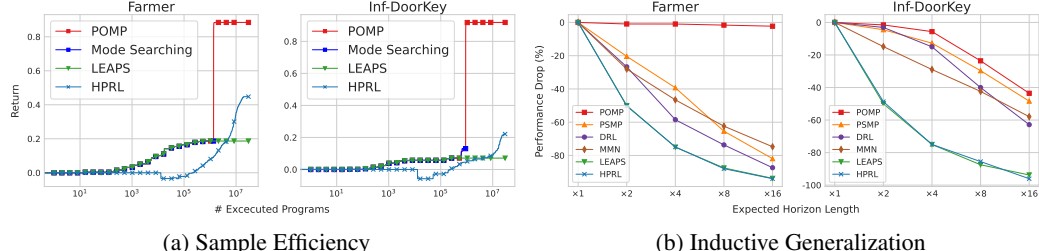

(a) Sample Efficiency          (b) Inductive Generalization

Figure 6: (a) **Program sample efficiency.** The training curves of POMP and other programmatic RL approaches, where the x-axis is the total number of executed programs for interacting with the environment, and the y-axis is the maximum validation return. This demonstrates that our proposed framework has better program sample efficiency and converges to better performance. (b) **Inductive generalization performance.** We evaluate and report the performance drop in the testing environments with an extended horizon, where the x-axis is the extended horizon length compared to the horizon of the training environments, and the y-axis is the performance drop in percentage. Our proposed framework can inductively generalize to longer horizons without any fine-tuning.

Furthermore, the state machine architecture of our approach provides not only the sustained execution of a singular skill but also the timely transition to another, as needed. Unlike the other tasks, SEESAW demands an extended traverse to obtain a marker, resulting in a more sparse reward distribution. During the search for mode programs, the emphasis on compatibility allows POMP to secure a set of mutually compatible modes that collaborate effectively to perform extended traversal. Some retrieved programs are shown in Figure 21, Figure 22 and Figure 23.

## 5.5 PROGRAM SAMPLE EFFICIENCY

To accurately evaluate the sample efficiency of programmatic RL methods, we propose the concept of *program sample efficiency*, which measures the total number of program executions required to learn a program policy. We report the program sample efficiency of LEAPS, HPRL, and POMP on FARMER and INF-DOORKEY, as shown in Figure 6a. POMP has better sample efficiency than LEAPS and HPRL, indicating that our framework requires less environmental interactions and computational cost. More details can be found in Section C.

## 5.6 INDUCTIVE GENERALIZATION

We aim to compare the inductive generalization ability of all the methods, which requires generalizing to instances requiring an arbitrary number of repetitions (Inala et al., 2020). To this end, we vary the horizons of FARMER and INF-DOORKEY and report the performance in Figure 6b. The results show that POMP experiences a smaller decline in performance in these testing environments with significantly extended horizons. This suggests that our approach exhibits superior inductive generalization in these tasks. Note that the longest execution of POMP runs up to 500k environment steps. More details about the tasks with extended horizons can be found in Section D.

## 6 CONCLUSION

This work aims to produce reinforcement learning policies that are human-interpretable and can inductively generalize by bridging program synthesis and state machines. To this end, we present the Program Machine Policy (POMP) framework for representing complex behaviors and addressing long-horizon tasks. Specifically, we introduce a method that can retrieve a set of effective, diverse, compatible programs by modifying the Cross Entropy Method (CEM). Then, we propose to use these programs as modes of a state machine and learn a transition function to transit among mode programs using reinforcement learning. To evaluate the ability to solve tasks with extended horizons, we design a set of tasks that requires thousands of steps in the Karel domain. Our framework POMP outperforms various deep RL and programmatic RL methods on the tasks. Also, POMP demonstrates superior performance in inductively generalizing to even longer horizons without fine-tuning. We conduct ablation studies that justify the effectiveness of our proposed search algorithm to retrieve mode programs and our proposed method to learn a transition function.

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

APPENDIX

## A    EXTENDED RELATED WORK

**Program Synthesis.** Program synthesis techniques revolve around program generation to convert given inputs into desired outputs. These methods have demonstrated notable successes across diverse domains such as array and tensor manipulation(Balog et al., 2017; Ellis et al., 2020), string transformation(Devlin et al., 2017; Hong et al., 2021; Zhong et al., 2023), generating computer command (Lin et al., 2018) and code (Chen et al., 2021; Li et al., 2022), graphics and 3D shape modeling(Wu et al., 2017; Liu et al., 2019; Tian et al., 2019), and describing agent behaviors(Bunel et al., 2018; Sun et al., 2018; Chen et al., 2019; Silver et al., 2020; Liang et al., 2022). Most program synthesis methods focus on task specifications such as input/output pairs, demonstrations, or language descriptions; in contrast, this work aims to synthesize human-readable programs as policies to solve reinforcement learning tasks.

**Hierarchical Reinforcement Learning and Semi-Markov Decision Processes.** HRL frameworks (Sutton et al., 1999; Barto & Mahadevan, 2003; Vezhnevets et al., 2017; Bacon et al., 2017) focus on learning and operating across different levels of temporal abstraction, enhancing the efficiency of learning and exploration, particularly in sparse-reward environments. In this work, Our proposed Program Machine Policy shares the same spirit and some ideas with HRL frameworks if we view the transition function as a "high-level" policy and the set of mode programs as "low-level" policies or skills. While most HRL frameworks either pre-define and pre-learned low-level policies, or jointly learn the high-level and low-level policies from scratch, our proposed framework first retrieves a set of effective, diverse, and compatible modes (*i.e.*, programs), and then learns the mode transition function.

The POMP framework also resembles the Option framework Sutton et al. (1999); Bacon et al. (2017); Klissarov & Precup (2021). More specifically, one can characterise POMP as using interpretable options as sub-policies since there is a high-level neural network being used to pick among retrieved programs as described in Section 4.3. Note that besides interpretable options, our work still differs from the option frameworks in the following aspects. Our work first retrieves a set of mode programs and then learns a transition function; this differs from most option frameworks that jointly learn options and a high-level policy that chooses options. Also, the transition function in our work learns to terminate, while the high-level policy option frameworks do not.

On the other hand, based on the definition of the recursive optimality described in Dietterich (1999), POMP can be categorized as recursively optimal since it is locally optimal given the policies of its children. Specifically, one can view the mode program retrieval process of POMP as solving a set of subtasks based on the proposed CEM-based search method that considers effectiveness, diversity, and compatibility. Then, POMP learns a transition function according to the retrieved programs, resulting in a policy as a whole. We have revised the paper to include this point of view.

**Symbolic Planning for Long-Horizon Tasks.** Another line of research uses symbolic operators (Yang et al., 2018; Guan et al., 2022; Cheng & Xu, 2023) for long-horizon planning. The major difference between POMP and Cheng & Xu (2023) and Guan et al. (2022) is the interpretability of the skill or option. In POMP, each learned skill is represented by a human-readable program. On the other hand, neural networks used in Cheng & Xu (2023) and tabular approaches used in Guan et al. (2022) are used to learn the skill policies. In Yang et al. (2018), the option set is assumed as input without learning and cannot be directly compared with Cheng & Xu (2023), Guan et al. (2022) and POMP.

Another difference between the proposed POMP framework, Cheng & Xu (2023), Guan et al. (2022), and Yang et al. (2018) is whether the high-level transition abstraction is provided as input. In Cheng & Xu (2023), a library of skill operators is taken as input and serves as the basis for skill learning. In Guan et al. (2022), the set of "landmarks" is taken as input to decompose the task into different combinations of subgoals. In PEORL (Yang et al., 2018), the option set is taken as input, and each option has a 1-1 mapping with each transition in the high-level planning. On the other hand, the proposed POMP framework utilized the set of retrieved programs as modes, which is conducted based on the reward from the target task without any guidance from framework input.

## B    DETAILS OF THE CROSS ENTROPY METHOD

### B.1    CEM

The pseudo-code of CEM is as follow:

---

**Algorithm 1** Cross Entropy Method

---

**procedure** CEM($G$, $g$, $\mathcal{P} = \mathcal{N}$, $N_s = 1000$, $n = 64$, $\sigma = 0.1$, $e = 0.1$)
    $z_r \leftarrow [z_0, z_1, ..., z_i, ..., z_{255}], z_i \sim \mathcal{P}$
    $step \leftarrow 0$
    **while** $step < N_s$ **do**
        $Z \leftarrow [\,]$
        $L_G \leftarrow [\,]$
        **for** $i \leftarrow 1$ to $n$ **do**
            $\varepsilon \leftarrow [\varepsilon_0, \varepsilon_1, ..., \varepsilon_i, ..., \varepsilon_{255}], \varepsilon_i \sim \mathcal{N}(0, \sigma_n)$
            $Z \leftarrow Z + [z_r + \varepsilon]$
            $L_G \leftarrow L_G + [G(Z[i-1], g)]$
        **end for**
        $R^{kl} \leftarrow \text{KthLargest}(L_G, n \cdot e)$
        $Z^{kl} \leftarrow [\,]$
        **for** $i \leftarrow 0$ to $n - 1$ **do**
            **if** $L_G[i] \leq R^{kl}$ **then**
                $Z^{kl} \leftarrow Z^{kl} + [Z[i]]$
            **end if**
        **end for**
        $z_r \leftarrow mean(Z^{kl})$
        $step \leftarrow step + 1$
    **end while**
**end procedure**

---

$G$ is the evaluation function, $g$ is the input of $G$, $\mathcal{P}$ is the distribution initial vector sampled from, $N_s$ is the maximum number of the iteration, $n$ is the population size, $\sigma$ is the standard deviation of the noise added to $z_r$, and $e$ is the percent of the population elites.

### B.2    CEM+DIVERSITY

The procedure of running CEM+diversity N times is as follows:

(1) Search the $1st$ program embedding $z_1$ by $CEM(G, g = (\{\} \cup Z_k, \Psi_{\text{before}}, \Psi_{\text{after}}))$

(2) Search the $2nd$ program embedding $z_2$ by $CEM(G, g = (\{z_1\} \cup Z_k, \Psi_{\text{before}}, \Psi_{\text{after}}))$

    ...

(N) Search the $Nth$ program embedding $z_N$ by $CEM(G, g = (\{z_1, ..., z_{N-1}\} \cup Z_k, \Psi_{\text{before}}, \Psi_{\text{after}}))$

In the simplest case, $Z_k$ is a empty set, $\Psi_{\text{before}}$ and $\Psi_{\text{after}}$ are empty sequences. It will then reduce to the process described in Section 4.2.2. An example of the searching trajectories can be seen in Figure 7.

### B.3    CEM+DIVERSITY+COMPATIBILITY

#### B.3.1    SAMPLE PROGRAM SEQUENCE

The procedure $H(z_1, z_2, z_3, ..., z_k)$ for sampling program embedding sequences $SQ_1$ and $SQ_2$ in Figure 3d from a total of $k$ programs is as follows:

(1) Randomly choose from the following options:

    a. With $(1 - \frac{1}{k+1})$ probability, sample a program embedding from $Z_y$ with replacement. Decode it to a program and append this program to $\Psi$.

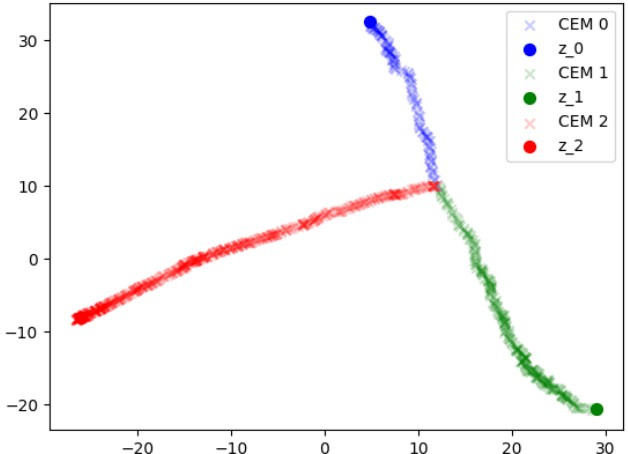

Figure 7: **CEM+Diversity Searching Trajectories.** It shows the trajectories of the procedure of running CEM+diversity 3 times. The program embeddings searched during the CEM are reduced to 2-dimensional embeddings using PCA. Since the diversity design, the $2nd$ CEM is forced to explore the opposite direction related to the searching path of the $1st$ CEM, and the $3rd$ CEM is urged to search a path that is perpendicular to the $1st$ and $2nd$ searching paths.

      b. With $\frac{1}{k+1}$ probability, stop sampling.

(2) Repeat (1) until the stop option is chosen.

After $SQ_1$ and $SQ_2$ are sampled, they are then be decoded to be $\Psi_{\text{before}}$ and $\Psi_{\text{after}}$, respectively.

### B.3.2 WHOLE PROCEDURE

The procedure of running CEM+diversity+Compatibility $|M|$ times in order to retrieve $|M|$ mode programs is as follows:

(1) Retrieve $1st$ mode program $z_1$.

      a. Run CEM+diversity N times with $Z_k = \{\}$, $\Psi_{\text{before}} = []$ and $\Psi_{\text{after}} = []$ to get N program embeddings.

      b. Choose the program embedding with the highest $G(z, \{\}, \Psi_{\text{before}}, \Psi_{\text{after}})$ among the N program embeddings as $z_1$.

(2) Retrieve $2nd$ mode program $z_2$.

      a. Sample $SQ_1$ and $SQ_2$ from $H(z_1)$, and deocde $SQ_1$ and $SQ_2$ to $\Psi_{\text{before}}$ and $\Psi_{\text{after}}$, respectively.

      b. Run CEM+diversity N times with $Z_k = \{z_1\}$, to get N program embeddings.

      c. Choose the program embedding with the highest $G(z, \{\}, \Psi_{\text{before}}, \Psi_{\text{after}})$ among the N program embeddings as $z_2$.

    ...

$(|M|)$ Retrieve $|M|th$ mode program $z_{|M|}$.

      a. Sample $SQ_1$ and $SQ_2$ from $H(z_1, z_2, ..., z_{|M|-1})$, and decode $SQ_1$ and $SQ_2$ to $\Psi_{\text{before}}$ and $\Psi_{\text{after}}$, respectively.

      b. Run CEM+diversity N times with $Z_k = \{z_1, z_2, ..., z_{|M|-1}\}$, to get N program embeddings.

      c. Choose the program embedding with the highest $G(z, \{\}, \Psi_{\text{before}}, \Psi_{\text{after}})$ among the N program embeddings as $z_{|M|}$.

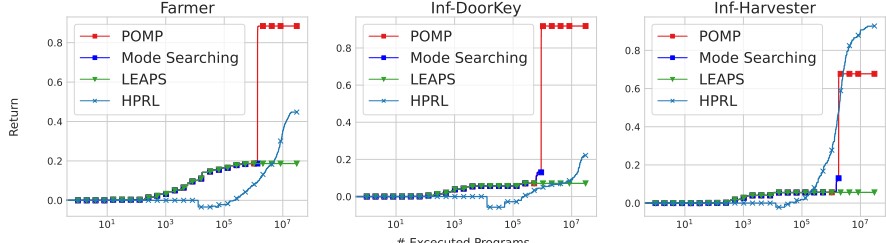

Figure 8: **Program Sample Efficiency.** Results of different programmatic RL approaches in FARMER, INF-DOORKEY, and INF-HARVESTER.

## C  PROGRAM SAMPLE EFFICIENCY

During the training of programmatic RL approaches, programs will be synthesized and executed in the environment of a given task to evaluate whether these programs are good enough. This three-step procedure (synthesis, execution, and evaluation) will be repeatedly done until the return converges or the maximum training steps are reached. The purpose of the analysis of the program sample efficiency is to figure out how many times this procedure needs to be done to achieve a certain return. As shown in Figure 8, POMP has the best program sample efficiencies in FARMER and INF-DOORKEY, but lower program sample efficiency than HPRL in INF-HARVESTER. The detail of the return calculation for each approach is described below.

### C.1  **POMP**

During the mode program searching process of POMP, a total of 265 CEMs are done for searching 5 mode programs. In each CEM, a maximum of 1000 iterations will be done, and in each iteration of the CEM, n (population size of the CEM) times of the three-step procedure are done. The return of a certain number of executed programs in the first half of the figure is recorded as the maximum return obtained from executing the previously searched programs solely.

During the transition function training process, the three-step procedure is done once in each PPO training step. The return of a certain number of executed programs in the remainder of the figure is recorded as the maximum validation return obtained by POMP.

### C.2  **LEAPS**

During the program searching process of LEAPS, the CEM is used for searching the program, and the hyperparameters of the CEM are tuned. A total of 216 CEMs are done. In each CEM, a maximum of 1000 iterations will be done, and in each iteration of the CEM, n (population size of the CEM) times of the three-step procedure are done. The return of a certain number of executed programs in the figure is recorded as the maximum return obtained from executing the previously searched programs solely.

### C.3  **HPRL**

During the meta-policy training process of HPRL, the three-step procedure is done once in each PPO training step. Therefore, with the setting of the experiment described in  Section F.6, as the training is finished, the three-step procedure will be done 25M times. The return of a certain number of executed programs in the figure is recorded as the maximum return obtained from the cascaded execution of 10 programs, which are decoded from latent programs output by the meta-policy.

## D  INDUCTIVE GENERALIZATION

To test the ability of the inductive generalization of different methods, we scale up the expected horizon of the environment by increasing the upper limit of the target for each KAREL-LONG task. To elaborate, using INF-DOORKEY as an example, the upper limit number of marker-picking and marker-placing is 16 under a training environment setting. Therefore, all policies are trained to

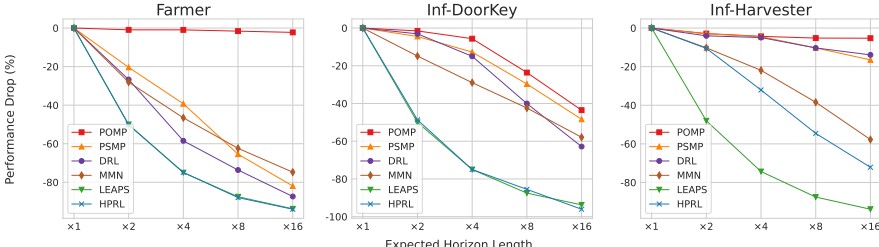

Figure 9: **Inductive Generalization.** Experiment Results on different baselines in FARMER, INF-DOORKEY, and INF-HARVESTER.

terminate after 16 markers are picked and placed. However, the upper limit number is set to 32, 64, etc, in the testing environment.

Since most of the baselines don't perform well on SEESAW and UP-N-DOWN, we do the inductive generalization experiments mainly on FARMER, INF-DOORKEY, and INF-HARVESTER. The expected horizon lengths of the testing environments will be 2, 4, 8, and 16 times longer compared to the training environment setting. Also, rewards gained from picking or placing markers and the penalty for actions are divided by 2, 4, 8, and 16 to normalize the maximum total reward of the tasks to 1. The detailed setting and the experiment result for each of these three tasks are shown as follows.

### D.1 FARMER

During the training phases of our and other baseline methods, we set the maximum iteration number to 2. However, we adjust this number for the testing phase to 4, 8, 16, and 32. As shown in Figure 9, when the expected horizon length grows, the performances of all the baselines except POMP drop dramatically, which means that our method has a much better inductive generalization property on this task.

### D.2 INF-DOORKEY

During the training phases of our and other baseline methods, we set the upper limit number of marker-picking and marker-placing to 16. However, we adjust this number for the testing phase to 32, 64, 128, and 256. As shown in Figure 9, when the expected horizon length grows, the performances of all the baselines drop considerably. Nevertheless, POMP has a minor performance drop compared to other baselines on this task.

### D.3 INF-HARVESTER

During the training phases of our and other baseline methods, we set the emerging probability to $\frac{1}{2}$. However, we adjust this number for the testing phase to $\frac{3}{4}$, $\frac{7}{8}$, $\frac{15}{16}$ and $\frac{31}{32}$. As shown in Figure 9, when the expected horizon length grows, the performances of POMP, PSMP, and DRL drop slightly, but the performances of MMN, LEAPS and HPRL drop extensively. Overall, POMP has a minimum performance drop among all baselines on this task.

## E  STATE MACHINE EXTRACTION

In our approach, since we employ the neural network transition function, the proposed program machine policies are only partially or locally interpretable – once the transition function selects a mode program, human users can read and understand the following execution of the program.

To further increase the interpretability of the trained mode transition function $f$, we extracted the state machine structure by the approach proposed in  (Koul et al., 2019). In this setup, since POMP utilizes the previous mode as one of the inputs and predicts the next mode, we focus solely on encoding the state observations. Each state observation is extracted by convolutional neural networks and fully connected layers to a $1 \times 128$ vector, which is then quantized into a $1 \times 5$ vector. We can construct a state-transition table using these quantized vectors and modes. The final step involves

minimizing these quantized vectors, which allows us to represent the structure of the state machine effectively. Examples of extracted state machine are shown in Figure 10, Figure 11 and Figure 12.

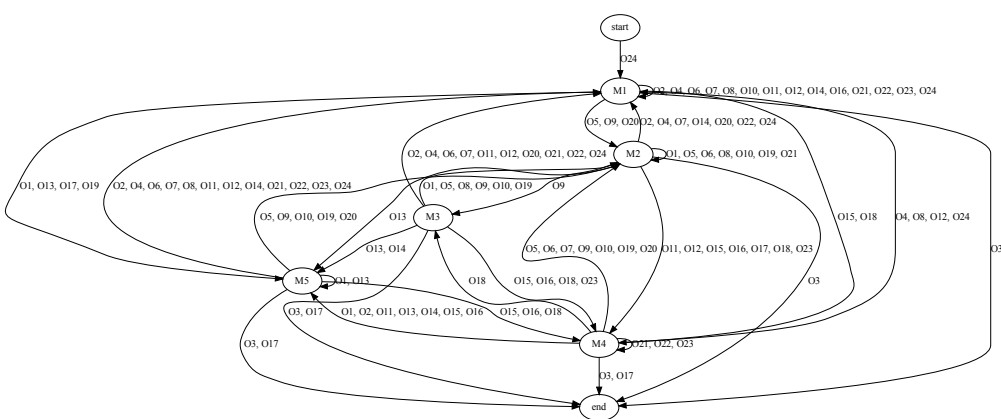

Figure 10: **Example of extracted state machine on FARMER**. $O1$ to $O24$ represent the unique quantized vectors encoded from observations. The corresponding mode programs of $M1$ to $M5$ are displayed in Figure 22.

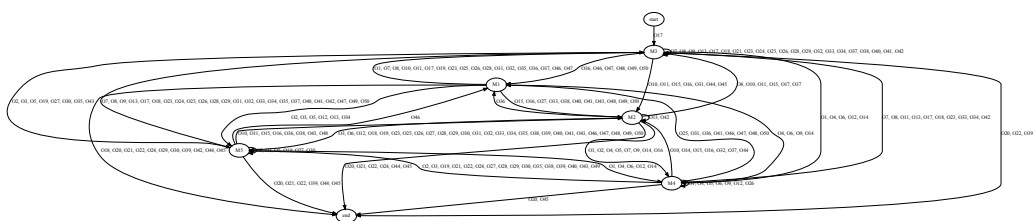

Figure 11: **Example of extracted state machine on INF-DOORKEY**. $O1$ to $O50$ represent the unique quantized vectors encoded from observations. The corresponding mode programs of $M1$ to $M5$ are displayed in Figure 22.

## F  HYPERPARAMETERS AND SETTINGS OF EXPERIMENTS

### F.1  POMP

**Encoder & Decoder.** We follow the training procedure and the model structure proposed in Trivedi et al. (2021), which uses recurrent networks to implement both the encoder $q_\phi$ and the decoder $p_\theta$ with hidden dimensions of 256 and trains them on programs randomly sampled from the Karel DSL. The model are updated through PPO (Schulman et al., 2017) algorithm and trained to optimize the $\beta$-VAE Higgins et al. (2016), the program behavior reconstruction loss, and the latent behavior reconstruction loss described in Trivedi et al. (2021).

The program dataset consists of 35,000 programs for training and 7,500 programs for validation and testing that were randomly sampled from the Karel DSL. We sequentially sample program tokens for each random program based on defined probabilities until an ending token or when a maximum program length is reached. The defined probability of each kind of token are listed below:

- WHILE: 0.15
- REPEAT: 0.03
- STMT_STMT: 0.5

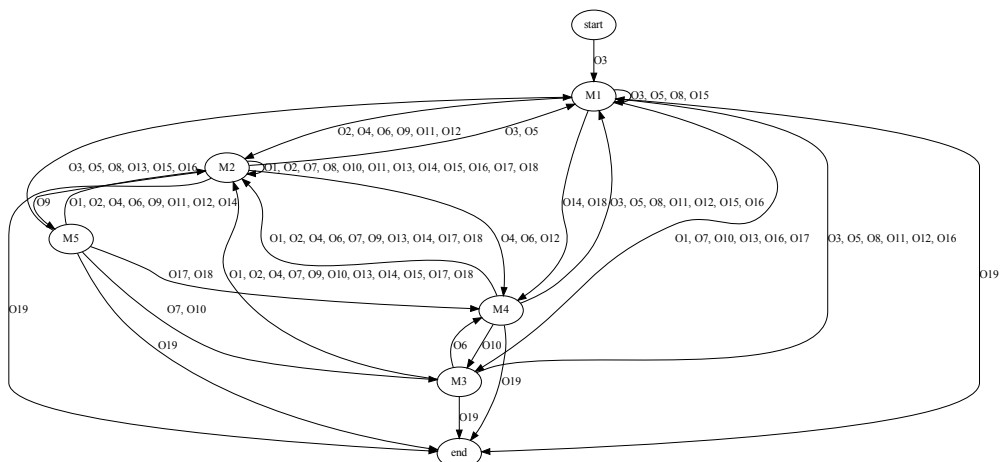

Figure 12: **Example of extracted state machine on INF-HARVESTER**. $O1$ to $O19$ represent the unique quantized vectors encoded from observations. The corresponding mode programs of $M1$ to $M5$ are displayed in Figure 23.

- `ACTION`: 0.2
- `IF`: 0.08
- `IFELSE`: 0.04

where `STMT_STMT` represents dividing the current token into two separate tokens, each chosen based on the same probabilities defined above. This token primarily dictates the length of the programs, as well as the quantity and complexity of nested loops and statements.

**Mode Program Synthesis.** To increase the diversity of the set of mode programs, we conduct 10 CEM searches in CEM+diversity to synthesize each mode program, and each program machine policy uses 5 modes from 5 CEM+diversity searches in the experiments. Only the hyperparameter set of the first CEM in the whole process is tuned. The rest of the CEMs use the same hyperparameter set as the first. The range of the hyperparameters is the same as Section F.5.

**Mode Transition Function.** The mode transition function $f$ consists of convolutional layers (Fukushima & Miyake, 1982; Krizhevsky et al., 2017) to derive features from the Karel states and the fully connected layers to predict the transition probabilities among each mode. Meanwhile, we utilize one-hot encoding to represent the current mode index of the program machine policy. The detail setting of the convolutional layers is the same as those described in Section F.4. The training process of the mode transition function $f$ can be optimized using the PPO (Schulman et al., 2017) algorithm. The hyperparameters are listed below:

- Maximum program number: 1000
- Batch size : 128
- Clipping: 0.05
- $\alpha$: 0.99
- $\gamma$: 0.99
- GAE lambda: 0.95
- Value function coefficient: 0.5
- Entropy coefficient: 0.1
- Number of updates per training iteration: 4
- Number of environment steps per set of training iterations: 32
- Number of parallel actors: 32

- Optimizer : Adam
- Learning rate: {0.1, 0.01, 0.001, 0.0001, 0.00001}

## F.2    PSMP

It resembles the setting in Section F.1. The input, output, and structure of the mode transition function $f$ remain the same. However, the 5 modes programs are replaced by 5 primitive actions (`move, turnLeft, turnRight, putMarker, pickMarker`).

## F.3    DRL

DRL training on the Karel environment uses the PPO (Schulman et al., 2017) algorithm with 20 million timesteps. Both the policies and value networks share a convolutional encoder that interprets the state of the grid world. This encoder comprises two layers: the initial layer has 32 filters, a kernel size of 4, and a stride of 1, while the subsequent layer has 32 filters, a kernel size of 2, and maintains the same stride of 1. The policies will predict the probability distribution of primitive actions (move, turnLeft, turnRight, putMarker, pickMarker) and termination. During our experiments with DRL on KAREL-LONG tasks, we fixed most of the hyperparameters and did hyperparameter grid search over learning rates. The hyperparameters are listed below:

- Maximum horizon: 10000
- Batch size : 128
- Clipping: 0.05
- $\alpha$: 0.99
- $\gamma$: 0.99
- GAE lambda: 0.95
- Value function coefficient: 0.5
- Entropy coefficient: 0.1
- Number of updates per training iteration: 4
- Number of environment steps per set of training iterations: 32
- Number of parallel actors: 32
- Optimizer : Adam
- Learning rate: {0.1, 0.01, 0.001, 0.0001, 0.00001}

## F.4    MMN

Aligned with the approach described in Koul et al. (2019), we trained and quantized a recurrent policy with a GRU cell and convolutional neural network layers to extract information from gird world states. During our experiments with MMN on KAREL-LONG tasks, we fixed most of the hyperparameters and did a hyperparameter grid search over learning rates. The hyperparameters are listed below:

- Hidden size of GRU cell: 128
- Number of quantized bottleneck units for observation: 128
- Number of quantized bottleneck units for hidden state: 16
- Maximum horizon: 10000
- Batch size : 128
- Clipping: 0.05
- $\alpha$: 0.99
- $\gamma$: 0.99
- GAE lambda: 0.95
- Value function coefficient: 0.5

- Entropy coefficient: 0.1
- Number of updates per training iteration: 4
- Number of environment steps per set of training iterations: 32
- Number of parallel actors: 32
- Optimizer : Adam
- Learning rate: {0.1, 0.01, 0.001, 0.0001, 0.00001}

## F.5 LEAPS

In line with the setup detailed in Trivedi et al. (2021), we conducted experiments over various hyperparameters of the CEM to optimize rewards for LEAPS. The hyperparameters are listed below:

- Population size (n): {8, 16, 32, 64}
- $\sigma$: {0.1, 0.25, 0.5}
- $e$: {0.05, 0.1, 0.2}
- Exponential $\sigma$ decay: {True, False}
- Initial distribution $\mathcal{P}$ : $\{\mathcal{N}(1,0), \mathcal{N}(0,\sigma), \mathcal{N}(0,0.1\sigma)\}$

## F.6 HPRL

In alignment with the approach described in Liu et al. (2023), we trained the meta policy for each task to predict a program sequence. To adapt this method to tasks with longer horizons, we increased the number of programs from 5 to 10. The hyperparameters are listed below:

- Max subprogram: 10
- Max subprogram Length: 40
- Batch size : 128
- Clipping: 0.05
- $\alpha$: 0.99
- $\gamma$: 0.99
- GAE lambda: 0.95
- Value function coefficient: 0.5
- Entropy coefficient: 0.1
- Number of updates per training iteration: 4
- Number of environment steps per set of training iterations: 32
- Number of parallel actors: 32
- Optimizer : Adam
- Learning rate: 0.00001
- Training steps: 25M

## G  DETAILS OF KAREL PROBLEM SET

The KAREL problem set is presented in Trivedi et al. (2021), consisting of the following tasks: STAIRCLIMBER, FOURCORNER, TOPOFF, MAZE, CLEANHOUSE and HARVESTER. Figure 13 and Figure 14 provide visual depictions of a randomly generated initial state, an internal state sampled from a legitimate trajectory, and the desired final state for each task. The experiment results presented in Table 1 and Table 3 are evaluated by averaging the rewards obtained from 32 randomly generated initial configurations of the environment.

### G.1 STAIRCLIMBER

This task takes place in a $12 \times 12$ grid environment, where the agent's objective is to successfully climb the stairs and reach the marked grid. The marked grid and the agent's initial location are both randomized at certain positions on the stairs, with the marked grid being placed on the higher end of the stairs. The reward is defined as $1$ if the agent reaches the goal in the environment, $-1$ if the agent moves off the stairs, and $0$ otherwise.

### G.2 FOURCORNER

This task takes place in a $12 \times 12$ grid environment, where the agent's objective is to place a marker at each of the four corners. The reward received by the agent will be $0$ if any marker is placed on the grid other than the four corners. Otherwise, the reward is calculated by multiplying $0.25$ by the number of corners where a marker is successfully placed.

### G.3 TOPOFF

This task takes place in a $12 \times 12$ grid environment, where the agent's objective is to place a marker on every spot where there's already a marker in the environment's bottom row. The agent should end up in the rightmost square of this row when the rollout concludes. The agent is rewarded for each consecutive correct placement until it either misses placing a marker where one already exists or places a marker in an empty grid on the bottom row.

### G.4 MAZE

This task takes place in an $8 \times 8$ grid environment, where the agent's objective is to find a marker by navigating the grid environment. The location of the marker, the initial location of the agent, and the configuration of the maze itself are all randomized. The reward is defined as $1$ if the agent successfully finds the marker in the environment, $0$ otherwise.

### G.5 CLEANHOUSE

This task takes place in a $14 \times 22$ grid environment, where the agent's objective is to collect as many scattered markers as possible. The initial location of the agent is fixed, and the positions of the scattered markers are randomized, with the additional condition that they will only randomly be scattered adjacent to some wall in the environment. The reward is defined as the ratio of the collected markers to the total number of markers initially placed in the grid environment.

### G.6 HARVESTER

This task takes place in an $8 \times 8$ grid environment, where the environment is initially populated with markers appearing in all grids. The agent's objective is to pick up a marker from each location within this grid environment. The reward is defined as the ratio of the picked markers to the total markers in the initial environment.

## H DETAILS OF KAREL-HARD PROBLEM SET

The KAREL-HARD problem set proposed by Liu et al. (2023) consists of the following tasks: DOORKEY, ONESTROKE, SEEDER and SNAKE. Each task in this benchmark is designed to have more constraints and be more structurally complex than tasks in the KAREL problem set. Figure 15 provides a visual depiction of a randomly generated initial state, some internal state(s) sampled from a legitimate trajectory, and the desired final state for each task. The experiment results presented in Table 1 and Table 3 are evaluated by averaging the rewards obtained from 32 randomly generated initial configurations of the environment.

### H.1 DOORKEY

This task takes place in an $8 \times 8$ grid environment, where the grid is partitioned into a $6 \times 3$ left room and a $6 \times 2$ right room. Initially, these two rooms are not connected. The agent's objective is to collect a key (marker) within the left room to unlock a door (make the two rooms connected) and

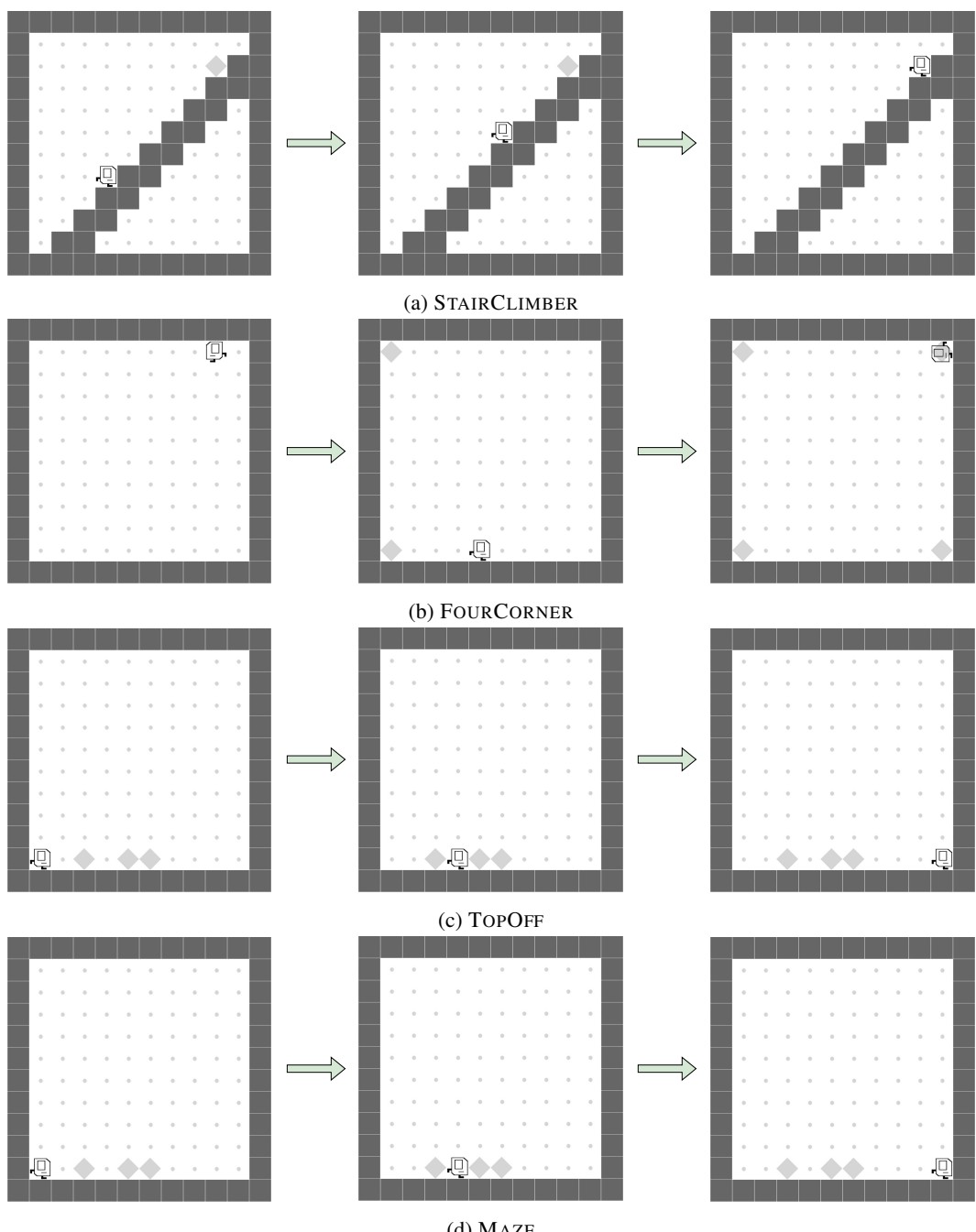

Figure 13: Visualization of STAIRCLIMBER, FOURCORNER, TOPOFF and MAZE in the KAREL problem set presented in Trivedi et al. (2021). For each task, a random initial state, a legitimate internal state, and the ideal end state are shown. In most tasks, the position of markers and the initial location of the Karel agent are randomized. More details of the KAREL problem set can be found in Section G.

precisely position the collected key atop a target (marker) situated in the right room subsequently. The agent's initial location, the key's location, and the target's location are all randomized. The agent receives a 0.5 reward for collecting the key and another 0.5 reward for putting the key on top of the target.

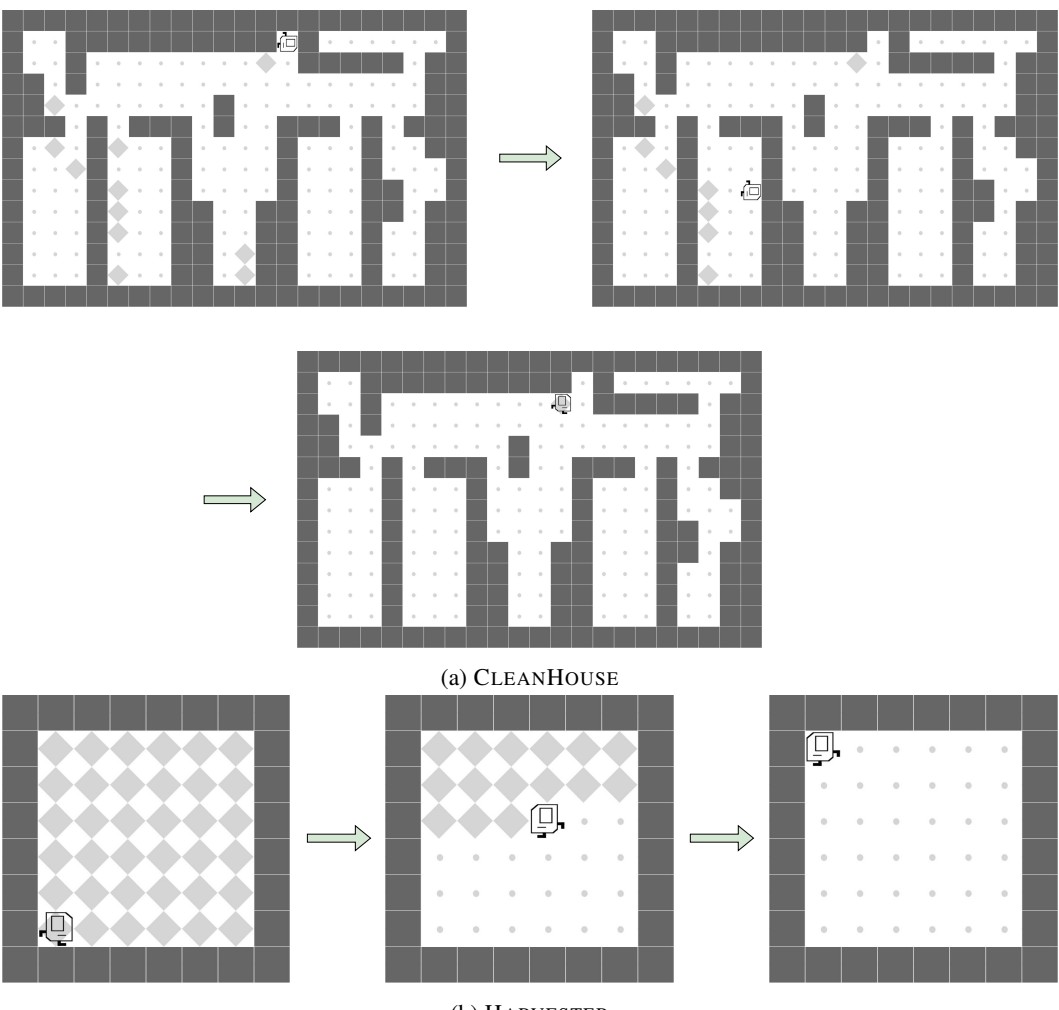

(a) CLEANHOUSE

(b) HARVESTER

Figure 14: Visualization of CLEANHOUSE and HARVESTER in the KAREL problem set presented in Trivedi et al. (2021). For each task, a random initial state, a legitimate internal state, and the ideal end state are shown. More details of the KAREL problem set can be found in Section G.

## H.2    ONESTROKE

This task takes place in an $8 \times 8$ grid environment, where the agent's objective is to navigate through all grid cells without revisiting any of them. Once a grid cell is visited, it transforms into a wall. If the agent ever collides with these walls, the episode ends. The reward is defined as the ratio of grids visited to the total number of empty grids in the initial environment.

## H.3    SEEDER

This task takes place in an $8 \times 8$ grid environment, where the agent's objective is to place a marker on every single grid. If the agent repeatedly puts markers on the same grid, the episode will then terminate. The reward is defined as the ratio of the number of markers successfully placed to the total number of empty grids in the initial environment.

## H.4    SNAKE

This task takes place in an $8 \times 8$ grid environment, where the agent plays the role of the snake's head and aims to consume (pass through) as much food (markers) as possible while avoiding colliding with its own body. Each time the agent consumes a marker, the snake's body length grows by 1, and

Table 3: **KAREL and KAREL-HARD Performance.** Mean return and standard deviation of all methods across the KAREL and KAREL-HARD problem set, evaluated over five random seeds. As the table shows, POMP outperforms LEAPS and HPRL on most KAREL and KAREL HARD tasks, except ONESTROKE and SNAKE, which need good long-term planning abilities. One thing to notice is that, unlike KAREL-LONG, KAREL and KAREL-HARD do not have per-action cost in design, so for POMP here, we did not let it learn to terminate but gave a maximum mode program execution number. Therefore, the whole policy will stop when this number is reached, or the task is solved.

| Method | Four Corner | Top Off | Clean House | Stair Climber | Harvester | Maze | Door Key | One Stroke | Seeder | Snake |
|---|---|---|---|---|---|---|---|---|---|---|
| DRL | $0.29 \pm 0.05$ | $0.32 \pm 0.07$ | $0.00 \pm 0.00$ | $1.00 \pm 0.00$ | $0.90 \pm 0.10$ | $1.00 \pm 0.00$ | $0.48 \pm 0.03$ | $\mathbf{0.89} \pm 0.04$ | $0.96 \pm 0.02$ | $\mathbf{0.67} \pm 0.17$ |
| LEAPS | $0.45 \pm 0.40$ | $0.81 \pm 0.07$ | $0.18 \pm 0.14$ | $1.00 \pm 0.00$ | $0.45 \pm 0.28$ | $1.00 \pm 0.00$ | $0.50 \pm 0.00$ | $0.65 \pm 0.19$ | $0.51 \pm 0.21$ | $0.21 \pm 0.15$ |
| HPRL (5) | $\mathbf{1.00} \pm 0.00$ | $\mathbf{1.00} \pm 0.00$ | $\mathbf{1.00} \pm 0.00$ | $1.00 \pm 0.00$ | $\mathbf{1.00} \pm 0.00$ | $1.00 \pm 0.00$ | $0.50 \pm 0.00$ | $0.80 \pm 0.02$ | $0.58 \pm 0.07$ | $0.28 \pm 0.11$ |
| POMP | $\mathbf{1.00} \pm 0.00$ | $\mathbf{1.00} \pm 0.00$ | $\mathbf{1.00} \pm 0.00$ | $1.00 \pm 0.00$ | $\mathbf{1.00} \pm 0.00$ | $1.00 \pm 0.00$ | $\mathbf{1.00} \pm 0.00$ | $0.62 \pm 0.01$ | $\mathbf{0.97} \pm 0.02$ | $0.36 \pm 0.02$ |

a new marker emerges at a different location. Before the agent successfully consumes 20 markers, there will consistently exist exactly one marker in the environment. The reward is defined as the ratio of the number of markers consumed by the agent to 20.

# I  DETAILS OF KAREL-LONG PROBLEM SET

Since none of the tasks in the KAREL and KAREL-HARD problem sets are truly long-horizon tasks, it is inadequate to use any of them as the environment when investigating the ability of our proposed framework. Hence, we introduce a newly designed KAREL-LONG problem set as a benchmark to evaluate the capability of POMP. Each task is designed to possess long-horizon properties based on the Karel states. Besides, we design the tasks in our KAREL-LONG benchmark to have a constant per-action cost (i.e., 0.0001). Figure 16, Figure 17, Figure 18, Figure 19 and Figure 20 provide visual depictions of all the tasks within the KAREL-LONG problem set. For each task, a randomly generated initial state and some internal states sampled from a legitimate trajectory are provided.

## I.1  EXPERIMENT RESULT DISCUSSION

As shown in Table 2, all approaches except ours struggle at the tasks in the KAREL-LONG problem set that require long-horizon exploration with sparse rewards (i.e., SEESAW and UP-N-DOWN). On the other hand, most approaches can achieve satisfactory performance on the tasks that provide dense rewards (i.e., FARMER, INF-DOORKEY, and INF-HARVESTER), even when the episode horizon is long.

In INF-DOORKEY, the environment consists of four small chambers (i.e., one 3x3, two 2x3, and one 2x2) connected. Therefore, each stage of INF-DOORKEY requires only moderate exploration compared to FARMER and INF-HARVESTER, allowing DRL to perform best. Besides, in INF-HARVESTER, the rewards provided are very dense, which is suitable for HPRL for optimizing the meta-policy and allowing it to perform best.

## I.2  SEESAW

This task takes place in a $16 \times 16$ grid environment, where the agent's objective is to move back and forth between two $4 \times 4$ chambers, namely the left chamber and the right chamber, to continuously collect markers. To facilitate movement between the left and right chambers, the agent must traverse through a middle $2 \times 6$ corridor. Initially, exactly one marker is randomly located in the left chamber, awaiting the agent to collect. Once a marker is picked in a particular chamber, another marker is then randomly popped out in the other chamber, further waiting for the agent to collect. Hence, the agent must navigate between the two chambers to pick markers continuously. The reward is defined as the ratio of the number of markers picked by the agent to the total number of markers that the environment is able to generate (emerging markers).

## I.3  UP-N-DOWN

This task takes place in an $8 \times 8$ grid environment, where the agent's objective is to ascend and descend the stairs repeatedly to collect markers (loads) appearing both above and below the stairs.

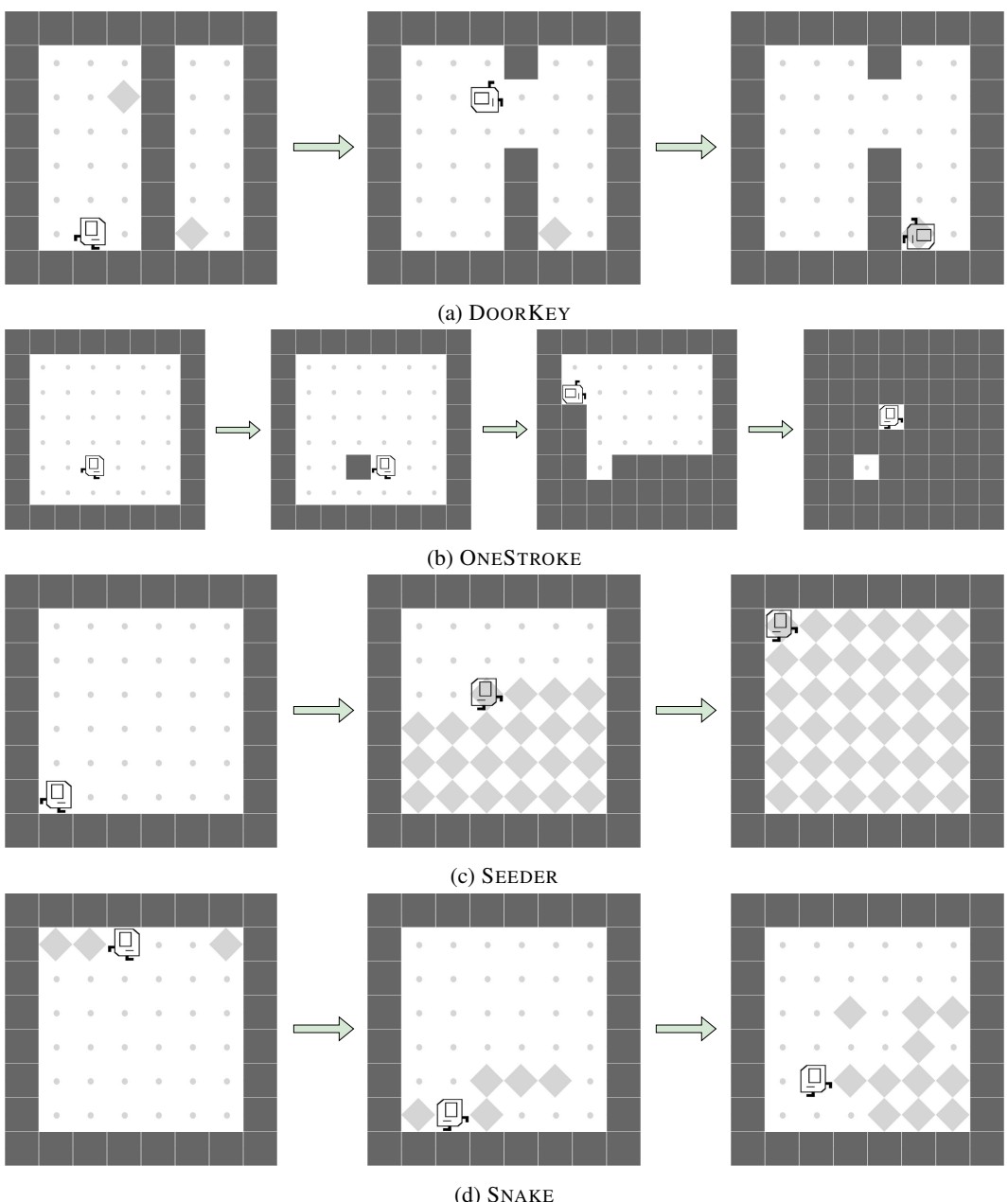

(a) DOORKEY

(b) ONESTROKE

(c) SEEDER

(d) SNAKE

Figure 15: Visualization of each task in the KAREL-HARD problem set proposed by Liu et al. (2023). For each task, a random initial state, some legitimate internal state(s), and the ideal end state are shown. More details of the KAREL-HARD problem set can be found in Section H.

Once a marker below (above) the stairs is picked up, another marker will appear above (below) the stairs, enabling the agent to continuously collect markers. If the agent moves to a grid other than those right near the stairs, the agent will receive a constant penalty (i.e., 0.005). The reward is defined as the ratio of the number of markers picked by the agent to the total number of markers that the environment is able to generate (emerging loads).

### I.4 FARMER

This task takes place in an $8 \times 8$ grid environment, where the agent's objective is to repeatedly fill the entire environment layout with markers and subsequently collect all of these markers. Initially,

all grids in the environment are empty except for the one in the upper-right corner. The marker in the upper-right corner is designed to be a signal that indicates the agent to start populating the environment layout with markers. After most of the grids are placed with markers, the agent is then asked to pick up markers as much as possible. Then, the agent is further asked to fill the environment again, and the whole process continues in this back-and-forth manner. We have set a maximum iteration number to represent the number of the filling-and-collecting rounds that we expect the agent to accomplish. The reward is defined as the ratio of the number of markers picked and placed by the agent to the total number of markers that the agent is theoretically able to pick and place (max markers).

### I.5  INF-DOORKEY

This task takes place in an $8 \times 8$ grid environment, where the agent's objective is to pick up a marker in certain chambers, place a marker in others, and continuously traverse between chambers until a predetermined upper limit number of marker-picking and marker-placing that we have set is reached. The entire environment is divided into four chambers, and the agent can only pick up (place) markers in one of these chambers. Once the agent does so, the passage to the next chamber opens, allowing the agent to proceed to the next chamber to conduct another placement (pick-up) action. The reward is defined as the ratio of markers picked and placed by the agent to the total number of markers that the agent can theoretically pick and place (max keys).

### I.6  INF-HARVESTER

This task takes place in a $16 \times 16$ grid environment, where the agent's objective is to continuously pick up markers until no markers are left and no more new markers are further popped out in the environment. Initially, the environment is entirely populated with markers. Whenever the agent picks up a marker from the environment, there is a certain probability (emerging probability) that a new marker will appear in a previously empty grid within the environment, allowing the agent to collect markers both continuously and indefinitely. The reward is defined as the ratio of the number of markers picked by the agent to the expected number of total markers that the environment can generate at a certain probability.

## J  DESIGNING DOMAIN-SPECIFIC LANGUAGES

Our program policies are designed to describe high-level task-solving procedures or decision-making logics of an agent. Therefore, our principle of designing domain-specific languages (DSLs) considers a general setting where an agent can perceive and interact with the environment to fulfill some tasks. DSLs consist of control flows, perceptions, and actions. While control flows are domain-independent, perceptions and actions can be designed based on the domain of interest, which would require specific expertise and domain knowledge.

Such DSLs are proposed and utilized in various domains, including ViZDoom (Kempka et al., 2016), 2D MineCraft (Sun et al., 2020), and gym-minigrid (Chevalier-Boisvert et al., 2023). Recent works (Liang et al., 2023; Wang et al., 2023) also explore describing agents' behaviors using programs with functions taking arguments.

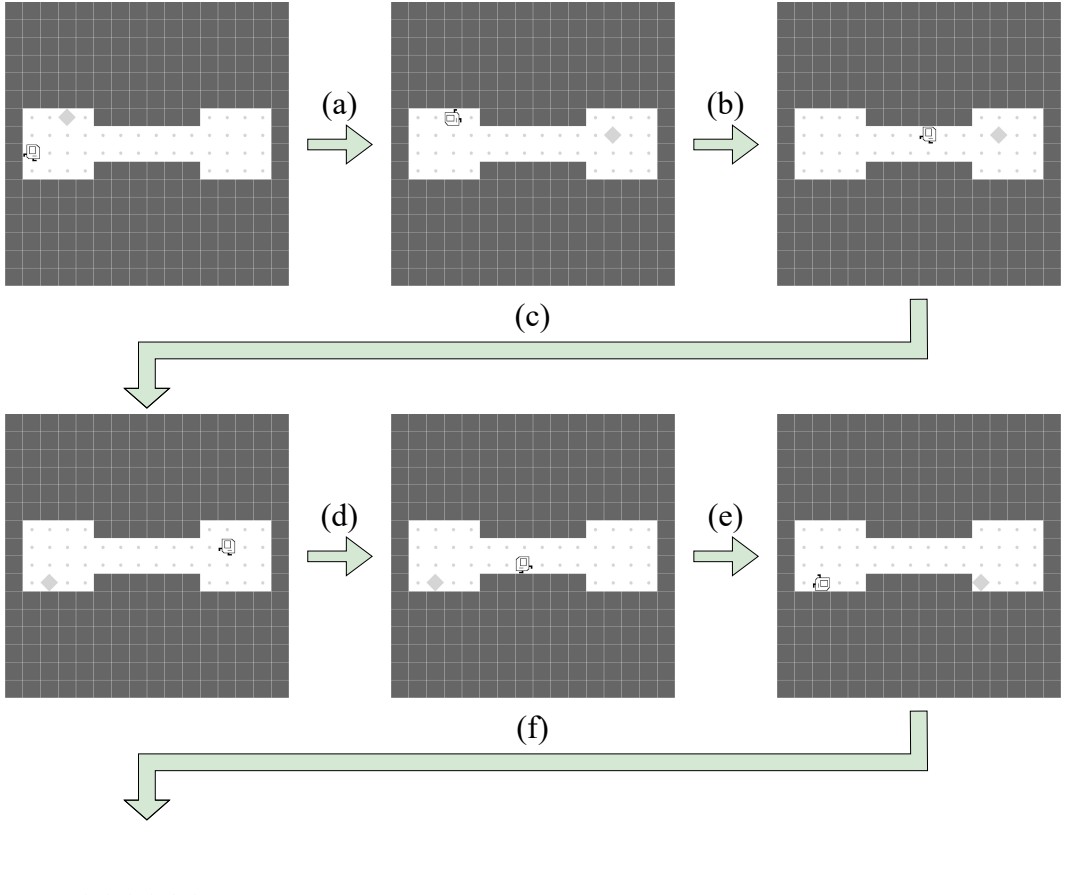

Figure 16: **Visualization of SEESAW in the KAREL-LONG problem set.** This partially shows a typical trajectory of the Karel agent during the task SEESAW. (a): Once the Karel agent collects a marker in the left chamber, a new marker will appear in the right chamber. (b): The agent must navigate through a middle corridor to collect the marker in the right chamber. (c): Upon the Karel agent collecting a marker in the right chamber, a new marker further appears in the left chamber. (d): Once again, the agent is traversing through the corridor to the left chamber. (e): A new marker appears in the right chamber again after the agent picks up the marker in the left chamber. (f): The agent will move back and forth between the two chambers to collect the emerging markers continuously. Note that the locations of all the emerging markers are randomized. Also, note that we have set the number of emerging markers to 10 during the training phase, meaning the agent has to pick up 10 markers to fully complete the task. More details of the task SEESAW can be found in Section I.

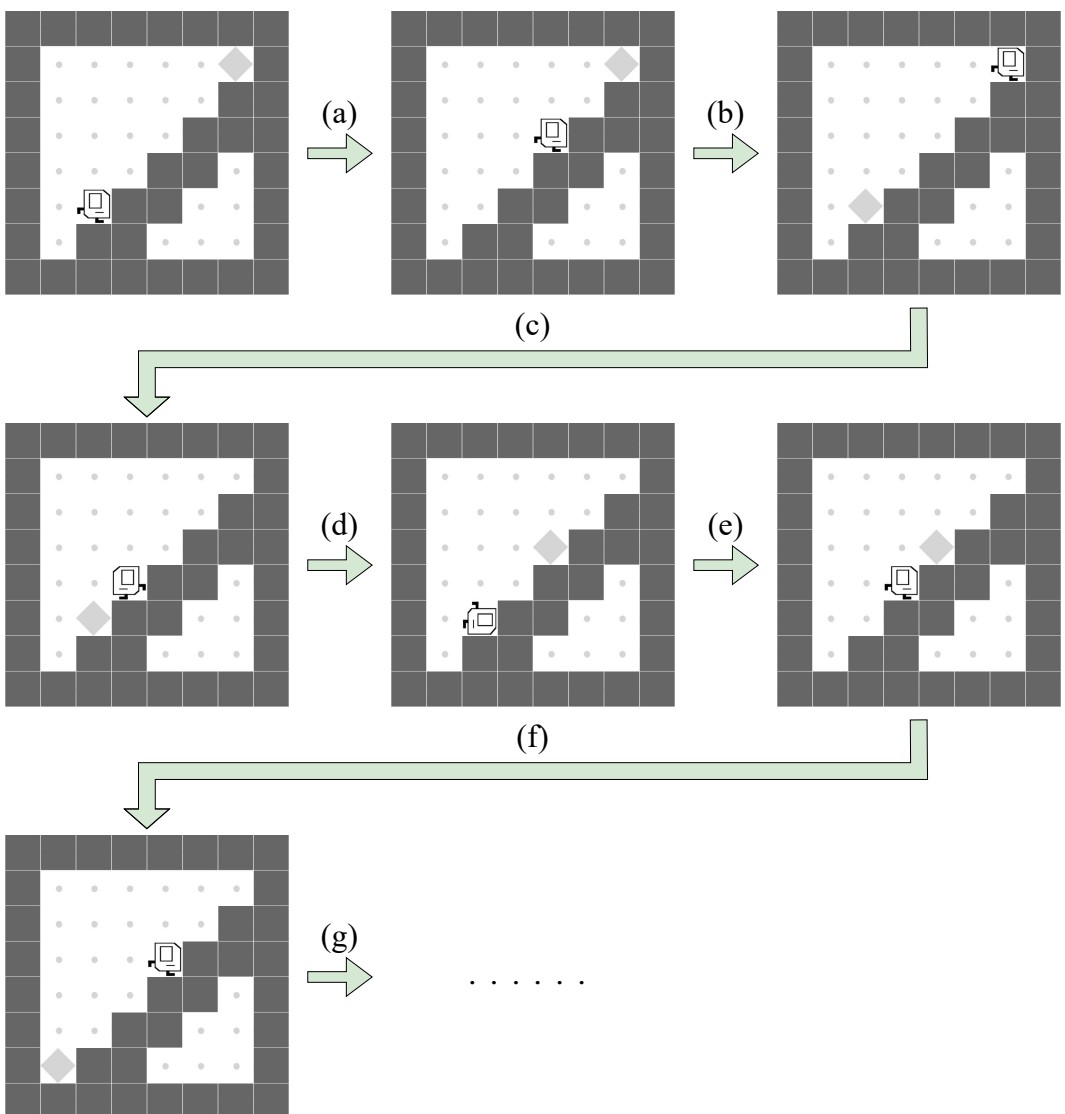

Figure 17: **Visualization of UP-N-DOWN in the KAREL-LONG problem set.** This partially shows a typical trajectory of the Karel agent during the task UP-N-DOWN. (a): The Karel agent is ascending the stairs to collect a load located above the stairs. Note that the agent can theoretically collect the load without directly climbing up the stairs, but it will receive some penalties if it does so. (b): Once the agent collects the load, a new load appears below the stairs. (c): The agent is descending the stairs to collect a load located below the stairs. Still, note that the agent can theoretically collect the load without directly climbing down the stairs, but it will receive some penalties if it does so. (d): Upon the agent collecting the load, a new load appears above the stairs. (e): The agent is again ascending the stairs to collect a load. (f): A new load appears below the stairs again after the agent collects the load located above the stairs. (g): The agent will then descend and ascend the stairs repeatedly to collect the emerging loads. Note that the locations of all the emerging loads are randomized right near the stairs, and they will always appear above or below the stairs depending on the position of the agent. Also, note that we have set the number of emerging loads to 10 during the training phase, meaning the agent has to collect 10 loads to fully complete the task. More details of the task UP-N-DOWN can be found in Section I.

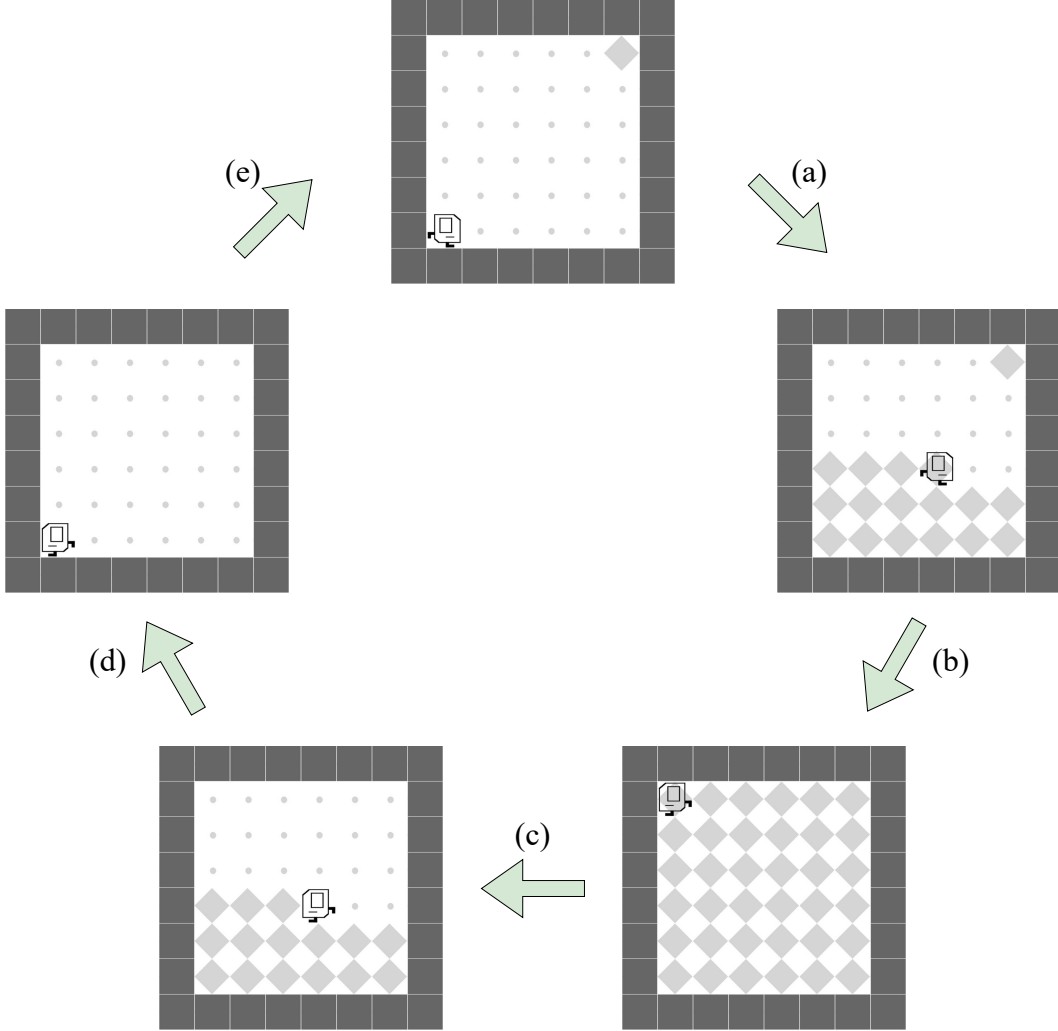

Figure 18: **Visualization of FARMER in the KAREL-LONG problem set.** This partially shows a typical trajectory of the Karel agent during the task FARMER. (a): The Karel agent is filling (placing) the entire environment layout with markers. Note that, in the initial state, there is a single marker located in the upper-right corner. The marker is designed to be a signal indicating the agent to start filling the environment layout. (b): The agent successfully populates the entire environment. (c): The agent is then asked to pick up markers as much as possible. (d): The agent successfully picks all markers up, leaving the environment empty. (e): If there is another filling-and-collecting round, a marker will appear in the upper-right corner to indicate that the agent should start the filling process again. Otherwise, the agent completes the entire task, and no further marker will appear. For simplicity, here, we only show the former case. Note that we have set the number of max markers to 144 during the training phase, meaning the agent has to both fill the entire environment layout with markers and pick up all markers twice to fully complete the task. More details of the task FARMER can be found in Section I.

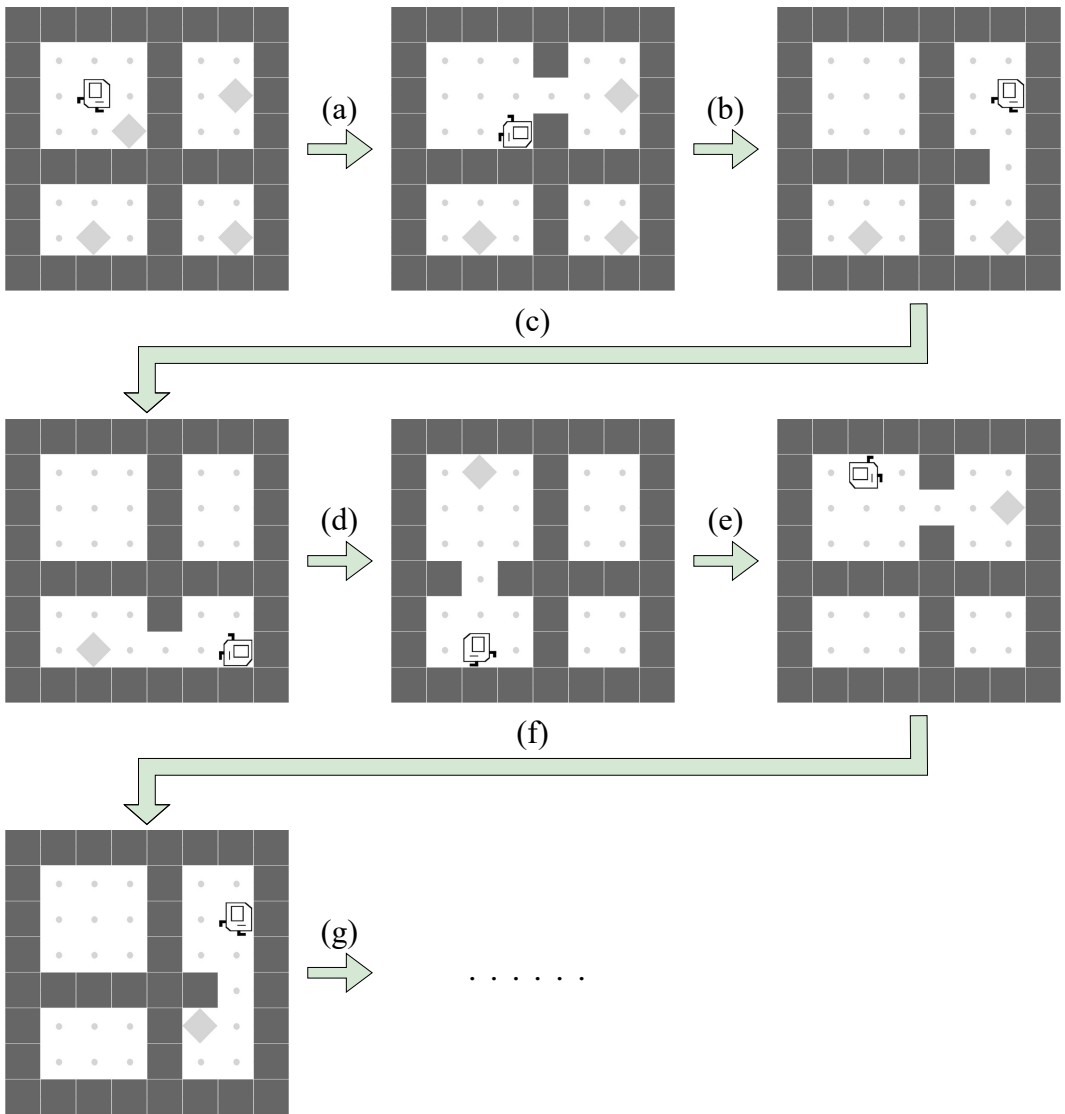

Figure 19: **Visualization of INF-DOORKEY in the KAREL-LONG problem set.** This partially shows a typical trajectory of the Karel agent during the task INF-DOORKEY. (a): The Karel agent picks up a marker in the upper-left chamber. Then, a passage to the upper-right chamber opens, allowing the agent to traverse through. (b): The agent successfully places a marker at a marked grid located in the upper-right chamber. Subsequently, a passage to the lower-right chamber opens, allowing the agent to traverse through. (c): After the agent collects a marker in the lower-right chamber, a passage to the lower-left chamber opens, allowing the agent to traverse through. (d): The agent properly places a marker at a marked grid located in the lower-left chamber. After that, a passage to the upper-left chamber opens and a new marker appears in the upper-left chamber. (e): Upon the agent picking up a marker in the upper-left chamber, the passage to the upper-right chamber opens again and a grid is marked randomly in the upper-right chamber. (f): The agent accurately places a marker at a marked grid located in the upper-right chamber. Afterward, the passage to the lower-right chamber opens again, and a new marker emerges in the lower-right chamber. (g): The agent will repeatedly pick up and place markers in this fashion until the number of max keys is reached. We have set the number of max keys to 16 during the training phase, meaning the agent has to pick up and place 16 markers in total to fully complete the task. More details of the task INF-DOORKEY can be found in Section I.

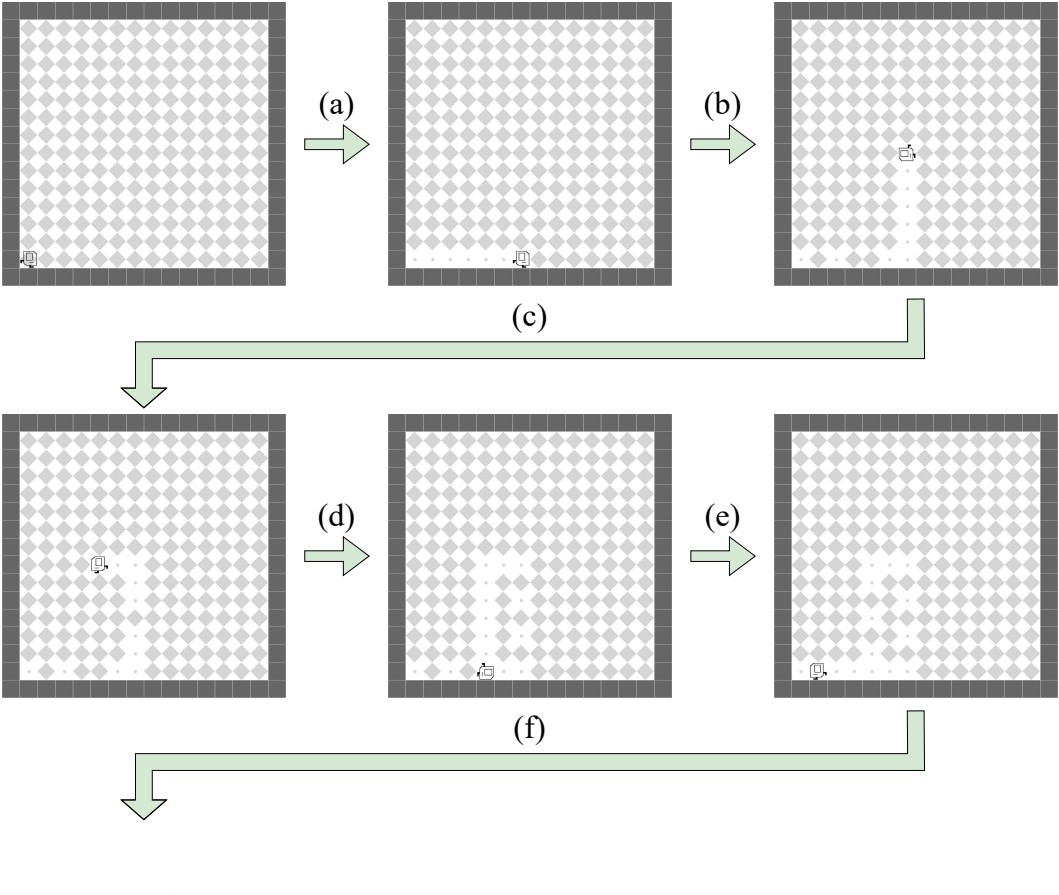

Figure 20: **Visualization of INF-HARVESTER in the KAREL-LONG problem set.** This partially shows a legitimate trajectory of the Karel agent during the task INF-HARVESTER. (a): The Karel agent is picking up markers in the last row. Meanwhile, no new markers are popped out in the last row. (b): The agent turns left and picks up 6 markers in the $7^{th}$ column. During this picking-up process, 3 markers appeared in 3 previously empty grids in the last row. (c): The agent is collecting markers in the $8^{th}$ row. During this picking-up process, 1 marker appeared in a previously empty grid in the $7^{th}$ column. (d): The agent picks up 6 markers in the $5^{th}$ column. During this picking-up process, 2 markers appeared in 2 previously empty grids in the $7^{th}$ column. (e): The agent picks up 2 more markers in the last row. During this picking-up process, 2 markers appeared in 2 previously empty grids in the $5^{th}$ column. (f): Since markers will appear in previously empty grids based on the emerging probability, the agent will continuously and indefinitely collect markers until no markers are left and no more new markers are further popped out in the environment. Note that we have set the emerging probability to $\frac{1}{2}$ during the training phase. More details of the task INF-HARVESTER can be found in Section I.

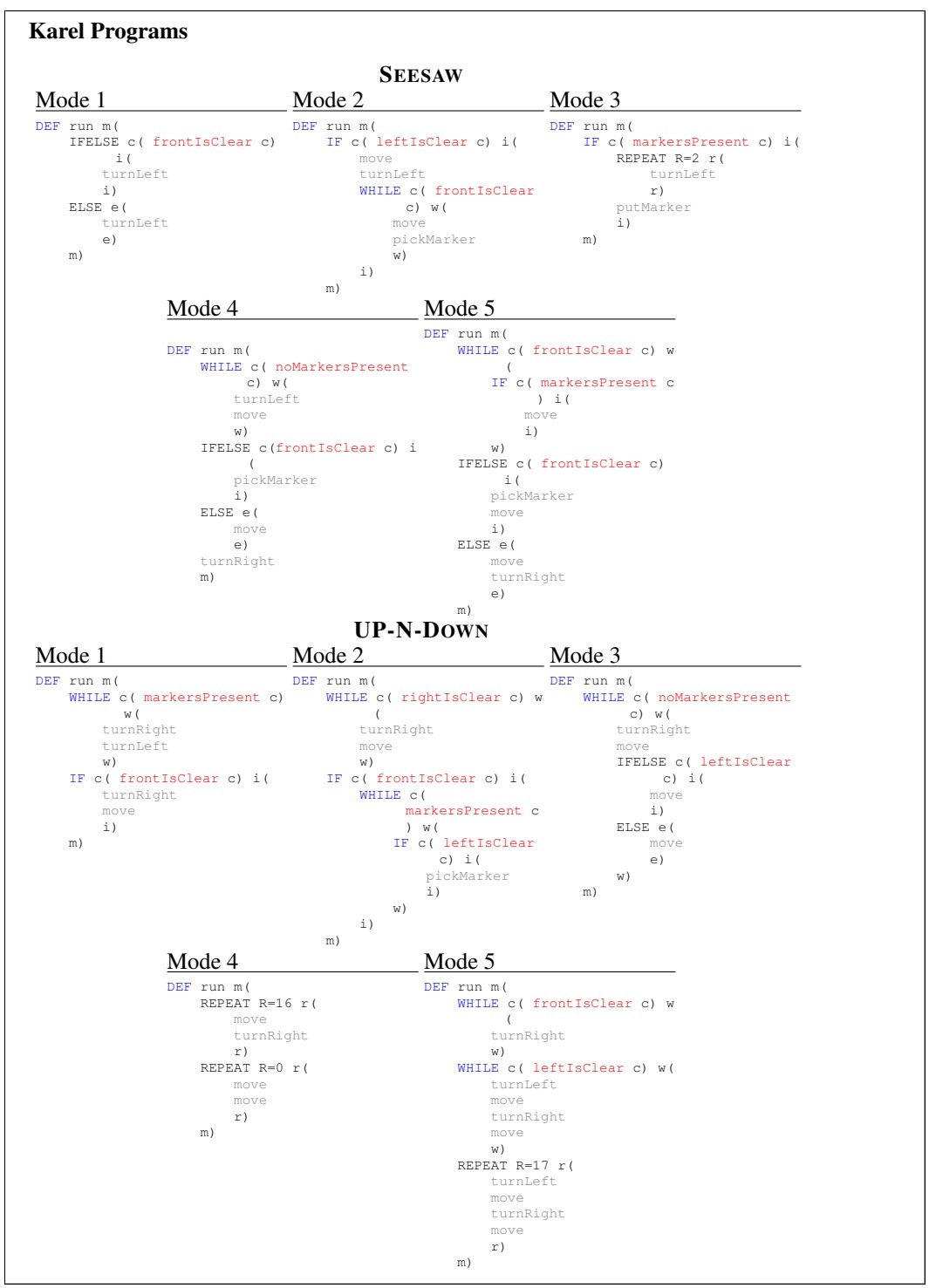

Figure 21: **Example programs on Karel-Long tasks: SEESAW and UP-N-DOWN.** The programs with best rewards out of all random seeds are shown.

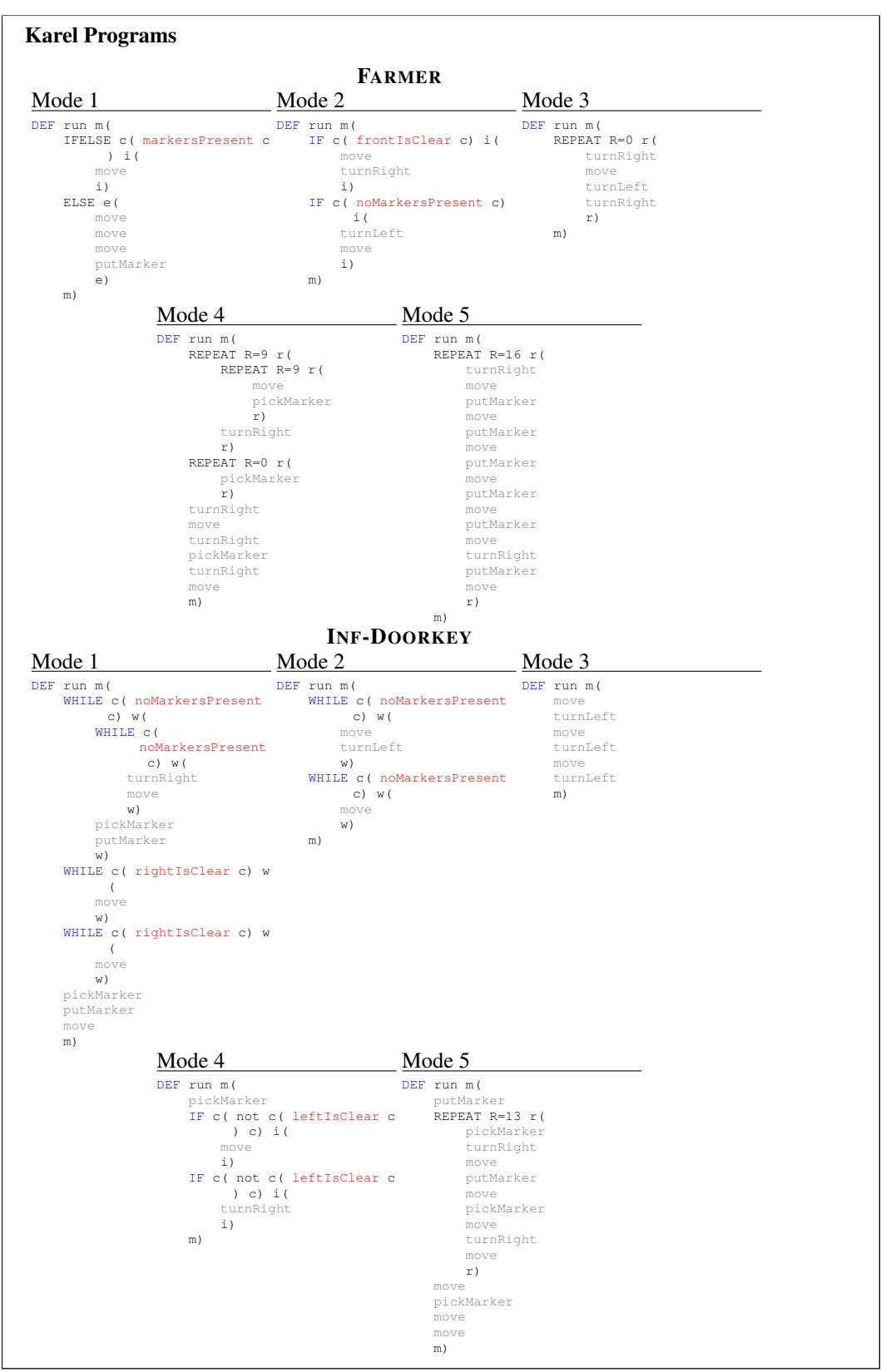

Figure 22: **Example programs on Karel-Long tasks: FARMER and INF-DOORKEY.** The programs with best rewards out of all random seeds are shown.

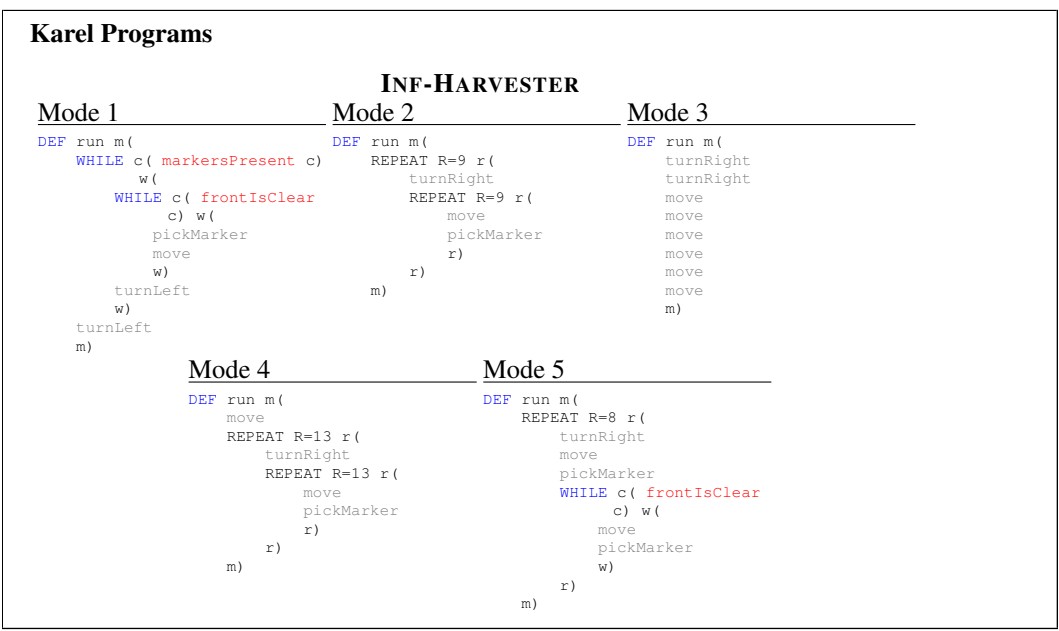

Figure 23: **Example programs on Karel-Long tasks: INF-HARVESTER.** The programs with best rewards out of all random seeds are shown.

