# OpenReview forum: "Addressing Long-Horizon Tasks by Integrating Program Synthesis and State Machines"
_ICLR.cc/2024/Conference — Submitted to ICLR 2024_

### Official Review · Reviewer_bA3y · 2023-10-23

**Soundness:** 2 fair
**Presentation:** 3 good
**Contribution:** 2 fair
**Rating:** 5
**Confidence:** 3

**Summary:**

The paper studies the problem of solving long-horizon tasks through programmatic RL. Specifically, the authors first learn the program embedding space for sampling the mode programs, and then learn a mode transition function to compose the programs towards the final task goal. Experiments are performed on several tabular tasks to validate the idea.

**Strengths:**

The paper presents an interesting idea that utilizes state machine representations and program generation for solving long-horizon tasks;

The performance is on par or better than other baselines from the evaluation perspective;

The paper is well-written and easily read.

**Weaknesses:**

The framework utilizes 3 stage design where the learning of the program embeddings and the mode transition function are separated, which could lead to cascading errors;

The absence of strict constraints to guarantee compatibility between generated programs could result in execution incompatibilities;

The experiments only consider simple tabular tasks, and only a few baselines are compared. Demonstrating the effectiveness of the framework on more complex tasks would strengthen the paper.

**Questions:**

In Sec. 4.2.3, the most compatible program is selected by inserting the newly generated program into a randomly chosen program sequence and ranking the returns of the execution, would this design guarantee compatibility? If not, what’s the failure ratio of the case that the generated program cannot be executed due to the compatibility issue?

It seems the method does not perform very ideally in the INF-DOORKEY and INF-HARVESTER, could authors provide additional explanations on this? Including more analysis of the failure modes would also help readers understand the limitations of the work;

Some related works also explore state machines for high-level transition abstraction in RL, e.g., integrating symbolic planning and skill policies [1][2][3], please consider discussing or comparing;

Reference:
[1] LEAGUE: Guided Skill Learning and Abstraction for Long-Horizon Manipulation, RA-L 2023;
[2] Leveraging approximate symbolic models for reinforcement learning via skill diversity, ICML 2022;
[3] PEORL: Integrating Symbolic Planning and Hierarchical Reinforcement Learning for Robust Decision-Making, IJCAI 2018.

---

> ### Author Response · Authors · 2023-11-22
> **Response to Reviewer bA3y (1/3)**
>
> We thank the reviewer for the thorough and constructive comments. Please find the response to your questions below.
>
> ### Responses to Questions
>
> > The framework utilizes 3 stage design where the learning of the program embeddings and the mode transition function are separated, which could lead to cascading errors.
>
> We thank the reviewer for raising this concern. In the following, we discuss how error propagation may happen from Stage 1 (i.e., learning a program embedding space) to Stage 2 (i.e., retrieving mode programs), and from Stage 2 to Stage 3 (i.e., constructing the policy by learning a transition function with the mode programs).
>
> **Stage 1 $\rightarrow$ Stage 2**: Stage 1 learns a program embedding space that continuously parameterizes diverse programs. Once the reconstruction loss is optimized, the program decoder can ideally decode all the training programs given corresponding program embeddings. In fact, the experiments show that with a sufficient embedding size, we can achieve 100% reconstruction accuracy on the entire training program set. Moreover, it can generate novel programs by interpolating in the learned program embedding space, as demonstrated in Trivedi et al., 2021. We believe the cascading error issue would not happen between Stage 1 and Stage 2, i.e., the training programs would not be "dropped/lost" in Stage 1.
>
> **Stage 2 $\rightarrow$ Stage 3**: Stage 2 retrieves a set of programs as modes of a state machine, and Stage 3 learns a transition function based on the retrieved mode programs. Therefore, as pointed out by the reviewer, if none of the programs retrieved from Stage 2 can solve a part of a task, Stage 3 cannot recover from such failure. To maximally avoid this issue of missing programs in Stage 2, this work focuses on designing search methods that can reliably retrieve all the essential programs in Stage 2. Specifically, we propose a CEM-based search method considering a set of programs' effectiveness, diversity, and compatibility. The ablation studies compare various methods for program retrieval and verify the superior performance of our proposed search method.
>
> > The absence of strict constraints to guarantee compatibility between generated programs could result in execution incompatibilities. In Sec. 4.2.3, the most compatible program is selected by inserting the newly generated program into a randomly chosen program sequence and ranking the returns of the execution, would this design guarantee compatibility? If not, what’s the failure ratio of the case that the generated program cannot be executed due to the compatibility issue?
>
> We thank the reviewer for raising this question. Each decoded program is guaranteed to be executable because a syntax mask is applied to the decoder during the program decoding process proposed by Trivedi et al., 2021 to ensure syntactic correctness. That said, we can safely (i.e., it does not “fail”) execute any programs in any order.
>
> We would like to clarify that we introduced the compatibility bonus to account for task performance instead of executability. For example, given a set of retrieved programs $p_1$ and $p_2$, the compatibility bonus tends to retrieve the third mode program $p_3$ if executing some orders (e.g., $p_1 \rightarrow p_3 \rightarrow p_2$,  $p_2 \rightarrow p_3$) of these three programs yields satisfactory rewards.
>
> > The experiments only consider simple tabular tasks, and only a few baselines are compared. Demonstrating the effectiveness of the framework on more complex tasks would strengthen the paper.
>
> **Baselines**: We believe that our comparisons to the baselines are sufficient. HPRL (Liu et al., 2023) is the current state-of-the-art method published at ICML 2023 (July 2023). We have revised the paper to include comparisons to the Moore machine network framework proposed by Koul et al., 2019, and state machine policies inspired by Inala et al., 2020. We would appreciate it if the reviewer could be specific about which methods we should compare against.
>
> **Tasks**: We would like to emphasize that the tasks in the Karel domain characterize various properties that resemble hard RL problems, such as long horizons (up to tens of thousands of time steps),  sparse rewards, deceptive rewards, etc. Nevertheless, we agree with the reviewer that extending our work to a variety of domains, e.g., ViZDoom (Kempka et al., 2016), 2D Minecraft (Sun et al., 2020), gym-minigrid (Chevalier-Boisvert et al., 2023), robot arm manipulation (Liang et al., 2023, Wang et al., 2023), is a promising research direction.

---

> ### Author Response · Authors · 2023-11-22
> **Response to Reviewer bA3y (2/3)**
>
> > It seems the method does not perform very ideally in the INF-DOORKEY and INF-HARVESTER, could authors provide additional explanations on this? Including more analysis of the failure modes would also help readers understand the limitations of the work.
>
> We thank the reviewer for raising this question.
>
> **Inf-Harvester**: We noted that in the Inf-Harvester task, there is a drastic difference between retrieving programs and learning the transition function. Specifically, when retrieving mode programs, it seldom starts executing program candidates from a state with a small number of markers, and therefore, the retrieved programs modes mainly are specialized in collecting “dense” markers; however, when learning the transition function, after executing a few programs, the state contains only a small number of markers, and no program mode is specialized in such a scenario.
>
> **Inf-DoorKey**: Our proposed POMP achieves the second-best performance in the Inf-DoorKey task, with an average reward of 0.91, outperforming HPRL (0.15) and LEAPS (0.1) by a large margin. DRL achieves the best performance (0.97). The Inf-DoorKey environment consists of four small chambers (i.e., one 3x3, two 2x3, and one 2x2) connected to each other. Therefore, each stage of Inf-Doorkey requires only moderate exploration, allowing DRL to achieve good performance.
>
> > Some related works also explore state machines for high-level transition abstraction in RL, e.g., integrating symbolic planning and skill policies [1][2][3], please consider discussing or comparing. Reference:
>
> > [1] LEAGUE: Guided Skill Learning and Abstraction for Long-Horizon Manipulation, RA-L 2023;
>
> > [2] Leveraging approximate symbolic models for reinforcement learning via skill diversity, ICML 2022;
>
> > [3] PEORL: Integrating Symbolic Planning and Hierarchical Reinforcement Learning for Robust Decision-Making, IJCAI 2018.
>
> We thank the reviewer for pointing out these relevant works on symbolic planning, and we have revised the paper to discuss them. The major difference between POMP and Cheng and Xu, 2023 and Lin et al., 2022 is the interpretability of the skill or option. In POMP, each learned skill is represented by a human-readable program. On the other hand, neural networks used by Cheng and Xu, 2023 and tabular approaches used by Lin et al., 2022 are used to learn the skill policies. In Yang et al., 2018, the option set is assumed as input without learning and cannot be directly compared with Cheng and Xu, 2023, Lin et al., 2022 and POMP.
>
> Another difference between the proposed POMP framework and Cheng and Xu, 2023, Lin et al., 2022, and Yang et al., 2018 is whether the high-level transition abstraction is provided as input. In Cheng and Xu, 2023, a library of skill operators is taken as input and serves as the basis for skill learning. In Lin et al., 2022, the set of “landmarks” is taken as input to decompose the task into different combinations of subgoals. In PEORL, the option set is taken as input, and each option has a 1-1 mapping with each transition in the high-level planning. On the other hand, the proposed POMP framework utilized the set of retrieved programs as  modes, which is conducted based on the reward from the target task without any guidance from framework input.

---

> ### Author Response · Authors · 2023-11-22
> **Response to Reviewer bA3y (3/3)**
>
> ### References
>
> - Liu et al. “Hierarchical Programmatic Reinforcement Learning via Learning to Compose Programs” ICML 2023
> - Inala et al. ”Synthesizing Programmatic Policies that Inductively Generalize” ICLR 2020
> - Trivedi et al. “Learning to Synthesize Programs as Interpretable and Generalizable Policies” NeurIPS 2021
> - Koul et al. "Learning Finite State Representations of Recurrent Policy Networks" in ICLR 2019
> - Kempka et al. "Vizdoom: A doom-based ai research platform for visual reinforcement learning" IEEE Conference on Computational Intelligence and Games 2016
> - Sun et al. ‘Program Guided Agent” in ICLR 2020
> - Chevalier-Boisvert, et al. “Minigrid & Miniworld: Modular & Customizable Reinforcement Learning Environments for Goal-Oriented Tasks’” arxiv 2023
> - Liang et al. “Code as Policies: Language Model Programs for Embodied Control” ICRA 2023
> - Wang et al. “Demo2Code: From Summarizing Demonstrations to Synthesizing Code via Extended Chain-of-Thought” arxiv 2023
> - Cheng and Xu "LEAGUE: Guided Skill Learning and Abstraction for Long-Horizon Manipulation" RA-L 2023
> - Lin et al. "Leveraging Approximate Symbolic Models for Reinforcement Learning via Skill Diversity" ICML 2022
> - Yang et al. "PEORL: Integrating Symbolic Planning and Hierarchical Reinforcement Learning for Robust Decision-Making" IJCAI 2018
>
> ### Conclusion
>
> We are incredibly grateful to the reviewer for the detailed and constructive review. We believe our responses address the concerns raised by the reviewer. Please kindly let us know if there are any further concerns or missing experimental results that potentially prevent you from accepting this submission. We would be more than happy to address them if time allows. Thank you very much for all your detailed feedback and the time you put into helping us to improve our submission.

---

### Official Review · Reviewer_dxYr · 2023-10-31

**Soundness:** 2 fair
**Presentation:** 3 good
**Contribution:** 3 good
**Rating:** 3
**Confidence:** 3

**Summary:**

This work proposes a programmatic RL framework which is both interpretable and capable of inductive generalisation. This is achieved by a three steps process. First the program embedding space is pre-trained using a VAE. This embedding space is then searched using CEM  to obtain programs which are then used to construct a finite state machine in the third step. The transitions between modes is also learned using RL. The primary technical contribution of this work lies particularly in how the programs are retrieved to ensure that they are able to obtain rewards (effective), provide new capabilities to the model (diverse) and also work well when applied in sequence (compatible).

**Strengths:**

## Originality
This work combines existing ideas in a unique way to achieve the proposed framework. The main originality of this work lies in the construction of the evaluation function which influences how the programs are obtained from the embedding space. Explicitly enforcing sampling of programs which are compatible with the previous and subsequent programs appears original and useful to the RL setting.

## Quality
This work clearly motivates each step in the pipeline and each component of the evaluation function. The figures of this paper are particularly high quality and helpful in presenting the pipeline. Overall the experimental design and setup is appropriate to evaluate the hypothesis of the work and the extension of the benchmarks is clearly able to determine the relative performance of the baseline algorithms and POMP, as well as the effect of the various ablations on POMP. Experimental results are interpreted fairly and presented clearly.

## Clarity
The work is written well and figures are legible and clear. An extremely minor point, in Figure 6a marker ticks aren't used but in Figure 6b they are. Consistency on this would be nice and if possible adding the ticks to 6a would be useful. Once again, I will note that the explanatory figures are particularly useful. The general consistency between them is makes them work together particularly well and presenting the ideas.

## Significance
Interpretability and hierarchical policies are both well established concepts and are important in RL. Thus, the prospect of having a hierarchical and interpretable model is definitely significant. I also agree that in most hierarchical frameworks it is usually the case that some clarity in how a policy achieves a certain goal is lost at the more detailed level. There is, however, a greater degree of clarity at the level of the hierarchical policy which transitions through the state machine or chooses programs/options/value functions. Thus, the proposed pipeline as it is presented would be clearly significant to the field for achieving an interpretable policy across both levels of the policy abstraction. Additionally, the new benchmarks appear challenging and increase the significance of the work. Overall, I can see this work leading to future work in programmatic RL, as well as being of practical use.

**Weaknesses:**

## Quality
My primary concern for this work is the depth to which the proposed pipeline is considered. I think that all explicit claims in this work are accurate, however I think that the pipeline is not challenged or considered fully. For example, the compatibility regulariser ensures that programs are compatible to the previous and subsequent programs. This does not seem to ensure compatibility across multiple programs. There is almost a markov assumption being implicitly used at the level of the programs. If this is true, it should be stated explicitly as it is an important point. It makes the comparison to Skill Machines less valid and also makes the pipeline resemble an Options framework far more. Moreover, skill machine are primarily used to encode long-horizon tasks where transitions between FSM states corresponds to a meaningful (usually semantic) change in the environment. However, since all all programs here are being sampled from a pre-trained embedding space which is not conditioned on the state or context of the environment, it seems all programs are only usable in the same environment, making them more monolithic in their own right. On the point of this resembling Options, how inaccurate would it be to characterise POMP as being interpretable Options since there is still a high-level neural network being used to pick programs.

Given the above my score is quite low to begin with, but I also acknowledge that some of my criticisms could also be due to me missing a subtle distinction between POMP and other works. So my confidence will also remain relatively low to start. I am certainly willing to increase both following a discussion on the above if it is shown that I have indeed not appreciated a point fully. I also reiterate, the work as proposed seems very significant, I am just not certain POMP goes all the way to achieving what is proposed.

Finally, I am concerned with the baselines which are being compared against. They seem to perform extremely poorly on the benchmarks, except in the cases where they work well and then they beat POMP. For example, DRL on the INF-DOORKEY domain and HPRL on INF_HARVESTER. While it beating POMP is not a problem, I would like to know what about this one particular domain made DRL work and better. Similarly in the ablation study, why are CEM and CEM+diversity capped at 0.5 performance. Why is the performance of CLEAN HOUSE and SNAKE particularly bad for both. Relatively little insight or discussion is given on these points. As a consequence it is difficult to get a good idea of the strength and weaknesses of POMP.

**Questions:**

I have asked a number of questions above in the weaknesses section which I think would aid my understanding greatly if answered. However, in line with some of what was mentioned there I would like to ask if the authors have a sense of the recursive optimality of POMP? This would also help contextualise or ground POMP in the rest of the hierarchical RL literature.

---

> ### Author Response · Authors · 2023-11-22
> **Response to Reviewer dxYr (1/4)**
>
> We thank the reviewer for the thorough and constructive comments. Please find the response to your questions below.
>
> ### Responses to Questions
>
> > An extremely minor point, in Figure 6a marker ticks aren't used but in Figure 6b they are. Consistency on this would be nice and if possible adding the ticks to 6a would be useful.
>
> We thank the reviewer for this suggestion. We have fixed it in the revised paper.
>
> > I think that the pipeline is not challenged or considered fully. For example, the compatibility regulariser ensures that programs are compatible to the previous and subsequent programs. This does not seem to ensure compatibility across multiple programs.
>
> We would like to emphasize that our proposed compatibility regulariser **ensures compatibility across the entire program sequence**. For example, say during the search phase, we have found 3 mode programs, $p_1$, $p_2$, $p_3$, and we are evaluating a candidate for the 4th program, $p_4$. At this iteration, if the program sequence $[p_2, p_1, p_4, p_3]$ was sampled, we will compute the execution reward by executing this entire program sequence, resulting in the compatibility score considering all $p_1$, $p_2$, $p_3$, not just the program executed right before and after $p_4$. We revised the paper to make this clear.
>
> As shown in Table 2, by taking this compatibility that considers entire program sequences described in Section 4.2.3 into consideration, CEM+diversity+compatibility outperforms other baselines that ignore the compatibility issues. This result highlights the effectiveness of the designed compatibility regulariser.
>
> > There is almost a markov assumption being implicitly used at the level of the programs. If this is true, it should be stated explicitly as it is an important point. It makes the comparison to Skill Machines less valid and also makes the pipeline resemble an Options framework far more.
>
> We would like to emphasize that we explicitly discuss this in the problem formulation section (Section 3) in the original paper. Specifically, it is stated that “the tasks considered in this work can be formulated as finite-horizon discounted Markov Decision Processes (MDPs).”
>
> We fully agree with the reviewer that our proposed Program Machine Policy shares the same spirit and some ideas with hierarchical reinforcement learning frameworks and semi-markov decision processes (SMDPs) if we view the transition function as a “high-level” policy and the set of mode programs as “low-level” policies or skills.
>
> While most HRL frameworks either pre-define and pre-learned low-level policies, or jointly learn the high-level and low-level policies from scratch, our proposed framework first retrieves a set of effective, diverse, and compatible modes, and then learns the mode transition function.
>
> We have revised the paper and included a discussion on HRL (Section A). We would appreciate it if the reviewer could suggest more related HRL and SMDP papers if our discussion misses anything.
>
> > Moreover, skill machine are primarily used to encode long-horizon tasks where transitions between FSM states corresponds to a meaningful (usually semantic) change in the environment. However, since all all programs here are being sampled from a pre-trained embedding space which is not conditioned on the state or context of the environment, it seems all programs are only usable in the same environment, making them more monolithic in their own right.
>
> We thank the reviewer for raising this question. Following Trivedi et al., 2021 and Liu et al., 2023, we learn the pre-trained program embedding space using randomly generated task-agnostic programs. Once a program embedding space of a domain (e.g., the Karel domain) has been learned, it can be used for solving any tasks in this domain. In fact, **we used the precisely same program embedding space to produce policies for all the tasks in the Karel, Karel-Hard, and Karel-Long problem sets**.
>
> Since we construct such a program embedding space from randomly generating a program dataset, the majority of the generated programs do not capture meaningful behaviors. Therefore, devising efficient search methods that can quickly identify effective programs is critical, which is the main focus of LEAPS and HRPL. The search method proposed in our work, on the other hand, not only aims to obtain effective programs but is also designed to retrieve a set of effective, diverse, and compatible programs. For example, Figure 7 illustrates how the proposed CEM+diversity can avoid searching over paths similar to those previously explored, which drastically increases the searching efficiency.

---

> > ### Author Response · Authors · 2023-11-22
> > **Response to Reviewer dxYr (2/4)**
> >
> > > On the point of this resembling Options, how inaccurate would it be to characterise POMP as being interpretable Options since there is still a high-level neural network being used to pick programs.
> >
> > We thank the reviewer for this insightful idea. This is exactly what our work does. We have revised the paper to include this (Section A). Note that besides interpretable options, our work still differs from the option frameworks in the following aspects. Our work first retrieves a set of mode programs and then learns a transition function; this differs from most option frameworks that jointly learn options and a high-level policy that chooses options. Also, the transition function in our work learns to terminate, while the high-level policy option frameworks do not.
> >
> > > I am concerned with the baselines which are being compared against. They seem to perform extremely poorly on the benchmarks, except in the cases where they work well and then they beat POMP. For example, DRL on the INF-DOORKEY domain and HPRL on INF_HARVESTER. While it beating POMP is not a problem, I would like to know what about this one particular domain made DRL work and better.
> >
> > We thank the reviewer for mentioning this observation. We discuss the performance of DRL and HPRL as follows and have included the discussions in the revised paper (Section I).
> >
> > DRL struggles at the tasks in the Karel-Long problem set that require long-horizon exploration with sparse rewards (i.e., Seesaw and Up-N-Down), as described in Section I. On the other hand, DRL can achieve satisfactory performance on the tasks which provide dense rewards (i.e., Farmer, Inf-DoorKey, and Inf-Harvester), even when the episode horizon is long.
> >
> > Particularly, the Inf-DoorKey task, the environment consists of four small chambers (i.e., one 3x3, two 2x3, and one 2x2) connected to each other. Therefore, each stage of Inf-Doorkey requires only moderate exploration than the whole task of Farmer and Inf-Harvester, allowing for DRL to achieve good performance.
> >
> > HPRL learns a meta-policy using RL and therefore can still struggle at long-horizon exploration with sparse rewards (i.e., Seesaw and Up-N-Down). However, HPRL learns to compose a fixed number of programs (in the original HPRL paper and 10 in our implementation), and hence still poorly on tasks with very long horizons (e.g., Inf-DoorKey). The rewards provided in the Inf-Harvester task are very dense, which make it easier to optimize the meta-policy in the HPRL framework.

---

> > > ### Author Response · Authors · 2023-11-22
> > > **Response to Reviewer dxYr (3/4)**
> > >
> > > > Similarly in the ablation study, why are CEM and CEM+diversity capped at 0.5 performance. Why is the performance of CLEAN HOUSE and SNAKE particularly bad for both. Relatively little insight or discussion is given on these points. As a consequence it is difficult to get a good idea of the strength and weaknesses of POMP.
> > >
> > > We thank the reviewer for raising this question.
> > >
> > > **DoorKey.** In the DoorKey task, the agent receives a reward of 0.5 once it picks up the marker (i.e., key) in the first room, and gets another reward of 0.5 when it places the marker at a desired location in the second room, as detailed in Section H. It is challenging because (1) It is impossible to reach the desired location before picking up the marker, which opens the door (i.e., wall) that separates these two rooms, and (2) any wrong marker placement causes a penalty.
> > >
> > > Since CEM and CEM+diversity only search for a single program, they only found the programs for picking up the marker in the first room, hence capping the performance at 0.5. On the other hand, our proposed POMP can solve the DoorKey task completely by using CEM+diversity+compatibility. The following two program modes are found by POMP.
> > >
> > > Program 1: Pick up the marker
> > > ```
> > > DEF run m( IF c( frontIsClear c) i( move i) turnRight WHILE c( frontIsClear c) w( pickMarker move w) m)
> > > ```
> > >
> > > Program 2: Place the marker at the desired location by trial and error.
> > > ```
> > > DEF run m(WHILE c( leftIsClear c) w( turnRight move putMarker pickMarker move w) m)
> > > ```
> > >
> > > **CleanHouse.** The minimum number of actions needed to solve CleanHouse completely is around 400, while the maximum number of actions a single program can trigger is about 200. Consequently, the programs found by CEM or CEM+diversity fail to achieve satisfactory performance, as shown in Table 1. CEM+diversity outperforms CEM, showing that our proposed diversity bonus can still lead to a better program.
> > >
> > > **Snake.** The Snake task immediately terminates if the agent hits its snake body or wall. Hence, solving this task requires long-term planning ability and is challenging to all the methods, as shown in Table 3. Nevertheless, as shown in Table 1, CEM+diversity outperforms CEM, showing that our proposed diversity bonus can still lead to a better program.
> > >
> > > As suggested by the reviewer, to provide an understanding of the strengths and weaknesses of POMP,  we have revised the paper to include the performance of DRL, LEAPS, HPRL, and our proposed framework on the Karel problem set and the Karel-Hard problem set in Table 3 (Section G and Section H). We also include the table in this response below to make it easier for the reviewer to read.
> > >
> > > | Method | FourCorner | TopOff | CleanHouse | StairClimber | Harvester | Maze | DoorKey | OneStroke | Seeder | Snake |
> > > |---|---|---|---|---|---|---|---|---|---|---|
> > > | **DRL** | 0.29 $\pm$ 0.05 | 0.32 $\pm$ 0.07 | 0.00 $\pm$ 0.00 | **1.00** $\pm$ 0.00 | 0.90 $\pm$ 0.10 | **1.00** $\pm$ 0.00 | 0.48 $\pm$ 0.03 | **0.89** $\pm$ 0.04 | 0.96 $\pm$ 0.02 | **0.67** $\pm$ 0.17 |
> > > | **LEAPS** | 0.45 $\pm$ 0.40 | 0.81 $\pm$ 0.07 | 0.18 $\pm$ 0.14 | **1.00** $\pm$ 0.00 | 0.45 $\pm$ 0.28 | **1.00** $\pm$ 0.00 | 0.50 $\pm$ 0.00 | 0.65 $\pm$ 0.19 | 0.51 $\pm$ 0.21 | 0.21 $\pm$ 0.15 |
> > > | **HPRL** | **1.00** $\pm$ 0.00 | **1.00** $\pm$ 0.00 | **1.00** $\pm$ 0.00 | **1.00** $\pm$ 0.00 | **1.00** $\pm$ 0.00 | **1.00** $\pm$ 0.00 | 0.50 $\pm$ 0.00 | 0.80 $\pm$ 0.02 | 0.58 $\pm$ 0.07 | 0.28 $\pm$ 0.11 |
> > > | **POMP** | **1.00** $\pm$ 0.00 | **1.00** $\pm$ 0.00 | **1.00** $\pm$ 0.00 | **1.00** $\pm$ 0.00 | **1.00** $\pm$ 0.00 | **1.00** $\pm$ 0.00 | **1.00** $\pm$ 0.00 | 0.62 $\pm$ 0.01 | **0.97** $\pm$ 0.02 | 0.36 $\pm$ 0.02 |
> > >
> > > We briefly summarize our findings as follows.
> > > - POMP can fully solve DoorKey by getting the key, opening the door, and placing the key at the desired location. DRL, LEAPS, and HPRL can only obtain the key.
> > > - POMP outperforms or performs competitively to DRL, LEAPS, and HPRL on 8 out of 10 tasks.
> > > - When POMP searches for the first mode program, the program retrieval process will be entirely equal to LEAPS, and POMP will take the exact computation cost to reach the same reward as LEAPS. On the other hand, HPRL requires more computational costs than POMP to earn the 1.0 reward on the Karel problem set. For DoorKey and Seeder, HPRL requires more computational costs than POMP to earn the same reward. For OneStroke and Snake, POMP requires more computational costs than HPRL to earn the same reward.

---

> > > > ### Author Response · Authors · 2023-11-22
> > > > **Response to Reviewer dxYr (4/4)**
> > > >
> > > > > if the authors have a sense of the recursive optimality of POMP? This would also help contextualise or ground POMP in the rest of the hierarchical RL literature.
> > > >
> > > > We thank the reviewer for providing this insightful view of hierarchical RL (HRL), which helps us contextualise POMP in the HRL landscape. Based on the definition of the recursive optimality described in Dietterich 2000, as pointed out by the reviewer, POMP can be categorized as recursively optimal since it is locally optimal given the policies of its children. Specifically, one can view the mode program retrieval process of POMP as solving a set of subtasks based on the proposed CEM-based  search method that considers effectiveness, diversity, and compatibility. Then, POMP learns a transition function according to the retrieved programs, resulting in a policy as a whole. We have revised the paper to include this point of view (Section A).
> > > >
> > > > ### References
> > > > - Trivedi et al. “Learning to Synthesize Programs as Interpretable and Generalizable Policies” in NeurIPS 2021
> > > > - Liu et al. “Hierarchical Programmatic Reinforcement Learning via Learning to Compose Programs” in ICML 2023
> > > > - Dietterich "Hierarchical reinforcement learning with the MAXQ value function decomposition" Artificial Intelligence Research 2000
> > > >
> > > > ### Conclusion
> > > >
> > > > We are incredibly grateful to the reviewer for the detailed and constructive review. We believe our responses address the concerns raised by the reviewer. Please kindly let us know if there are any further concerns or missing experimental results that potentially prevent you from accepting this submission. We would be more than happy to address them if time allows. Thank you very much for all your detailed feedback and the time you put into helping us to improve our submission.

---

### Official Review · Reviewer_HDvE · 2023-11-02

**Soundness:** 2 fair
**Presentation:** 2 fair
**Contribution:** 1 poor
**Rating:** 5
**Confidence:** 5

**Summary:**

The paper introduces a system that combines program synthesis and finite state machines to generate policies for solving reinforcement learning problems. The authors built on the previous work of LEAPS, where one trains a latent space of programs that is used as a search space for policies. The LEAPS space is used to generate a set of diverse behaviors (modes), and later, a reinforcement learning algorithm is trained to learn how to transition from one mode to the next.

The system is evaluated on novel Karel tasks, which includes the need of repetitions.

**Strengths:**

The problem of learning policies with programmatic representations is an important one and this paper makes a contribution in this line of research. I also enjoyed the idea of mixing synthesis in the Karel language with finite state machines. Although I find it a little odd the dichotomy used in the paper: programmatic and FSM. I find it odd because a FSM is also a program, but in a language different from the Karel language. Perhaps the main contribution of this work is to show how to combine the inductive biases of two languages, where one is used to learn small functions and the other is used to combine such functions.

**Weaknesses:**

Although I like many of the ideas in the paper and I also found it very easy to read and understand, I have several important issues with the current submission.

**Interpretability Motivation**

The main motivation of the work is around interpretability. The paper states already in the abstract the separation between FSM and programmatic solutions, where the former allows for repetitive behaviors and the latter for interpretability. The paper specifically cites the work of Inala et al. as being difficult to be interpreted by human users. Similarly to how I can accept that programs in the Karel language can be interpretable, I can also accept that the FSM learned with Inala et al.'s system to also be interpretable. As far as I know, no paper has actually evaluated the interpretability of these systems, so the argument that FSM aren't interpretable is wrong to me.

Even if we accept that the policies the system of Inala et al. generates aren't interpretable and the programs in the Karel language are, I still find it hard to accept that the policies POMP encodes are interpretable. The choices of which mode to execute next are provided by a neural network, which arguably is hard to interpret. While it is easy to accept that the modes are interpretable (although no evidence for it was provided), I would accept that the POMP policies are interpretable only after seeing some strong evidence of the fact.

**Empirical Methodology**

I apologize if I missed this in the paper, but is the process of learning modes accounted for in the curves shown in Figure 6a and in the numbers shown in Table 2? And is the learning of modes performed in the target task? I ask this question because, due to the compatible constrain, learning modes is essentially learning a solution to the problem because it already tries to combine programs such that they can maximize the expected reward. I might have also missed this information, but it isn't clear the number of steps used to train each system.

It is disappointing that POMP was not evaluated on the original Karel problems from LEAPS and HPRL. I was particularly interested in seeing POMP performance in the original DoorKey problem. Since both CEM and CEM+diversity get the reward of 0.50 with 0.0 standard deviation, I can only assume that the system learns how to get the key, but it doesn't open the door. Due to the way POMP is trained, it would not find a mode that knows how to act once the agent picks up the key. I suspect POMP would take much longer to reach the same reward of LEAPS and HPRL. I would like to see results of POMP on the original set of problems, to see how it compares with LEAPS and HPRL.

Since I was not convinced with the interpretability argument against FSM and in favor of POMP, the experiments miss a baseline that learns FSMs. If not Inala et al.'s method, then the method by Koul et al. (Learning Finite State Representations Recurrent Policy Networks).

I also missed a baseline that searches directly in the space of programs, like in genetic programming. For example, Aleixo & Lelis (Show Me the Way! Bilevel Search for Synthesizing Programmatic Strategies) use a two-level search with simulated annealing to search for programmatic policies. How would this method compare?

What about NDS by Verma et al. (Programmatically interpretable reinforcement learning), can it be used to find programs for Karel?

**Questions:**

1. Is the process of learning modes accounted for in the curves shown in Figure 6a and in the numbers shown in Table 2?

2. Is the learning of modes performed in the target task?

3. How does POMP perform in the original DoorKey domain?

4. Why not consider other baselines such as NDS and Simulated Annealing?

---

> ### Author Response · Authors · 2023-11-22
> **Response to Reviewer HDvE (1/3)**
>
> We thank the reviewer for the thorough and constructive comments. Please find the response to your questions below.
>
> ### Responses to Questions
>
> > The paper specifically cites the work of Inala et al. as being difficult to be interpreted by human users. Similarly to how I can accept that programs in the Karel language can be interpretable, I can also accept that the FSM learned with Inala et al.'s system to also be interpretable. As far as I know, no paper has actually evaluated the interpretability of these systems, so the argument that FSM aren't interpretable is wrong to me.
>
> We thank the reviewer for pointing that out. We agree with the reviewer that symbolic programs used in Inala et al., 2020's systems are more interpretable than neural network policies. However, we still believe that such symbolic mode programs describing the relationship among variables, which are presented in Figure 16 to Figure 21 in Inala et al., 2020 paper, are less interpretable than our programs written in a formal language. Trivedi et al., 2021 demonstrated that users with basic programming skills can interpret, edit, and improve the programs synthesized by LEAPS, which highlights the interpretability of DSL programs.
>
> As suggested by the reviewer, we have revised the paper and toned down our statement. Instead of stating that the FSMs used in Inala et al., 2020 are "difficult to be interpreted by human users," we now argue such FSMs "are less interpretable (compared to DSL programs)."
>
> > Even if we accept that the policies the system of Inala et al. generates aren't interpretable and the programs in the Karel language are, I still find it hard to accept that the policies POMP encodes are interpretable. The choices of which mode to execute next are provided by a neural network, which arguably is hard to interpret. While it is easy to accept that the modes are interpretable (although no evidence for it was provided), I would accept that the POMP policies are interpretable only after seeing some strong evidence of the fact.
>
> We agree with the reviewer that employing the neural network transition function makes our proposed program machine policies not entirely interpretably. However, the program machine policies still enjoy "local interpretibility." That said, once the transition function chooses each mode program, human users can read and understand the following execution of the program. To reflect this point raised by the reviewer, we have revised the paper to emphasize that the policies produced by our proposed framework are "locally interpretable." We appreciate the reviewer for helping us clarify this.
>
> To further improve the interpretability of the proposed program machine policies, we have conducted additional experiments to extract Moore machines from learned transition functions using the Moore machine network framework proposed by Koul et al., 2019. By synthesizing the corresponding Moore machine from a learned transition function, we can explicitly display the learned mode transition mechanism as a state machine and thus make the overall framework more interpretable. The extracted results on "Farmer," "Inf-Doorkey," and "Inf-Harvester" are shown in Section E of the revised paper.
>
> > Is the process of learning modes accounted for in the curves shown in Figure 6a and in the numbers shown in Table 2? And is the learning of modes performed in the target task? I ask this question because, due to the compatible constrain, learning modes is essentially learning a solution to the problem because it already tries to combine programs such that they can maximize the expected reward. I might have also missed this information, but it isn't clear the number of steps used to train each system.
>
> We thank the reviewer for raising the questions.
> - Yes, timesteps of learning modes are accounted for in the curves shown in Figure 6a. The detail about how returns in Figure 6a are calculated is addressed in Section C of the revised paper.
> - Yes, the mode program retrieving process introduced in Section 4.2.3 is performed on each target task to ensure that the retrieved mode programs are diverse and compatible for solving the given task.

---

> > ### Author Response · Authors · 2023-11-22
> > **Response to Reviewer HDvE (2/3)**
> >
> > > How does POMP perform in the original DoorKey domain?
> > It is disappointing that POMP was not evaluated on the original Karel problems from LEAPS and HPRL. I was particularly interested in seeing POMP performance in the original DoorKey problem. Since both CEM and CEM+diversity get the reward of 0.50 with 0.0 standard deviation, I can only assume that the system learns how to get the key, but it doesn't open the door. Due to the way POMP is trained, it would not find a mode that knows how to act once the agent picks up the key. I suspect POMP would take much longer to reach the same reward of LEAPS and HPRL. I would like to see results of POMP on the original set of problems, to see how it compares with LEAPS and HPRL.
> >
> > We have revised the paper to include the performance of DRL, LEAPS, HPRL, and our proposed framework on the Karel problem set and the Karel-Hard problem set in Table 3 (Section G and Section H). We also include the table in this response below to make it easier for the reviewer to read.
> >
> > | Method | FourCorner | TopOff | CleanHouse | StairClimber | Harvester | Maze | DoorKey | OneStroke | Seeder | Snake |
> > |---|---|---|---|---|---|---|---|---|---|---|
> > | **DRL** | 0.29 $\pm$ 0.05 | 0.32 $\pm$ 0.07 | 0.00 $\pm$ 0.00 | **1.00** $\pm$ 0.00 | 0.90 $\pm$ 0.10 | **1.00** $\pm$ 0.00 | 0.48 $\pm$ 0.03 | **0.89** $\pm$ 0.04 | 0.96 $\pm$ 0.02 | **0.67** $\pm$ 0.17 |
> > | **LEAPS** | 0.45 $\pm$ 0.40 | 0.81 $\pm$ 0.07 | 0.18 $\pm$ 0.14 | **1.00** $\pm$ 0.00 | 0.45 $\pm$ 0.28 | **1.00** $\pm$ 0.00 | 0.50 $\pm$ 0.00 | 0.65 $\pm$ 0.19 | 0.51 $\pm$ 0.21 | 0.21 $\pm$ 0.15 |
> > | **HPRL** | **1.00** $\pm$ 0.00 | **1.00** $\pm$ 0.00 | **1.00** $\pm$ 0.00 | **1.00** $\pm$ 0.00 | **1.00** $\pm$ 0.00 | **1.00** $\pm$ 0.00 | 0.50 $\pm$ 0.00 | 0.80 $\pm$ 0.02 | 0.58 $\pm$ 0.07 | 0.28 $\pm$ 0.11 |
> > | **POMP** | **1.00** $\pm$ 0.00 | **1.00** $\pm$ 0.00 | **1.00** $\pm$ 0.00 | **1.00** $\pm$ 0.00 | **1.00** $\pm$ 0.00 | **1.00** $\pm$ 0.00 | **1.00** $\pm$ 0.00 | 0.62 $\pm$ 0.01 | **0.97** $\pm$ 0.02 | 0.36 $\pm$ 0.02 |
> >
> > We briefly summarize our findings as follows.
> > - POMP can fully solve DoorKey by getting the key, opening the door, and placing the key at the desired location. DRL, LEAPS, and HPRL can only obtain the key. Specifically, POMP learns to use the following program to solve the second stage of the DoorKey task, trying every grid to find the correct grid to place the key.
> > ```
> > DEF run m(WHILE c( leftIsClear c) w( turnRight move putMarker pickMarker move w) m)
> > ```
> > - POMP outperforms or performs competitively to DRL, LEAPS, and HPRL on 8 out of 10 tasks.
> > - When POMP searches for the first mode program, the program retrieval process will be entirely equal to LEAPS, and POMP will take the exact computation cost to reach the same reward as LEAPS. On the other hand, HPRL requires more computational costs than POMP to earn the 1.0 reward on the Karel problem set. For DoorKey and Seeder, HPRL requires more computational costs than POMP to earn the same reward. For OneStroke and Snake, POMP requires more computational costs than HPRL to earn the same reward.
> >
> > > Since I was not convinced with the interpretability argument against FSM and in favor of POMP, the experiments miss a baseline that learns FSMs. If not Inala et al.'s method, then the method by Koul et al. (Learning Finite State Representations Recurrent Policy Networks).
> >
> > We thank the reviewer for this suggestion. We agree that including comparisons to methods that produce state machine policies will strengthen our work. Therefore, we have implemented and evaluated a state machine policy, dubbed programmatic state machine policy (PSMP), whose modes are primitive actions of the agent, instead of mode programs retrieved by our method, resembling the idea of Inala et al., 2020. Also, we have implemented and evaluated the Moore machine network (MMN; Koul et al., 2019). The results are reported in Table 2 and discussed in Section 5.4 of the revised paper. We have also included the results below.
> >
> > | Method | Seesaw | UP-N-Down | Farmer | Inf-DoorKey | Inf-Harvester |
> > |---|---|---|---|---|---|
> > | **PSMP** | 0.25 $\pm$ 0.39 | 0.00 $\pm$ 0.00 | 0.58 $\pm$ 0.06 | **0.97** $\pm$ 0.01 | 0.59 $\pm$ 0.06 |
> > | **MMN** | 0.10 $\pm$ 0.00 | 0.00 $\pm$ 0.00 | 0.57 $\pm$ 0.12 | 0.87 $\pm$ 0.05 | 0.41 $\pm$ 0.10 |
> > | **POMP** | **0.90** $\pm$ 0.02 | **0.97** $\pm$ 0.00 | **0.88** $\pm$ 0.01 | 0.91$\pm$ 0.01 | **0.67** $\pm$ 0.03 |
> >
> > As shown above, our proposed POMP consistently outperforms MMN across all the tasks and achieves better performance than PSMP on 4 out of 5 tasks by large margins. More details of these two methods and their performance can be found in the revised paper. We sincerely thank the reviewer for this suggestion, which helps us improve our work.

---

> > > ### Author Response · Authors · 2023-11-22
> > > **Response to Reviewer HDvE (3/3)**
> > >
> > > > I also missed a baseline that searches directly in the space of programs, like in genetic programming. For example, Aleixo & Lelis (Show Me the Way! Bilevel Search for Synthesizing Programmatic Strategies) use a two-level search with simulated annealing to search for programmatic policies. How would this method compare? Why not consider other baselines such as NDS (Verma et al. (Programmatically interpretable reinforcement learning)) and Simulated Annealing?
> > >
> > > We thank the reviewer for suggesting these works. However, Bi-S proposed in Aleixo & Lelis, 2023 and Neurally Directed Program Search (NDPS; Verma et al., 2018) require feature functions as input or hand-crafted program sketches, which differs from our experimental settings that only produce programmatic policies from reward signals. We have revised the paper to include a discussion on Aleixo & Lelis, 2023, as the original paper has referred to Verma et al., 2018.
> > >
> > > ### References
> > >
> > > - Inala et al. ”Synthesizing Programmatic Policies that Inductively Generalize” in ICLR 2020
> > > - Trivedi et al. “Learning to Synthesize Programs as Interpretable and Generalizable Policies” in NeurIPS 2021
> > > - Koul et al. "Learning Finite State Representations of Recurrent Policy Networks" in ICLR 2019
> > > - Aleixo and Lelis "Show Me the Way! Bilevel Search for Synthesizing Programmatic Strategies" in AAAI 2023
> > > - Verma et al. "Programmatically Interpretable Reinforcement Learning" in ICML 2018
> > >
> > > ### Conclusion
> > >
> > > We are incredibly grateful to the reviewer for the detailed and constructive review. We believe our responses address the concerns raised by the reviewer. Please kindly let us know if there are any further concerns or missing experimental results that potentially prevent you from accepting this submission. We would be more than happy to address them if time allows. Thank you very much for all your detailed feedback and the time you put into helping us to improve our submission.

---

### Official Review · Reviewer_NAAH · 2023-11-02

**Soundness:** 3 good
**Presentation:** 2 fair
**Contribution:** 2 fair
**Rating:** 5
**Confidence:** 3

**Summary:**

This work proposes to solve long-horizon tasks by synthesizing programs to accomplish the task. Since existing methods for synthesizing programs struggle with long-horizon tasks, this work proposes to synthesize simple programs and then learn to compose these programs. Specifically, this work takes an approach involving three steps:
(1) An embedding space over programs is learned.
(2) A set of diverse and reusable programs is retrieved by searching over the embedding space.
(3) A function is learned to compose these programs: i.e., to choose which programs to execute one after another.

This work evaluates its proposed approach on a benchmark of Karel tasks, involving picking and placing objects in a grid world and finds that the proposed approach performs favorably compared to prior approaches for program synthesis.

**Strengths:**

This work investigates an interesting and novel method for solving RL tasks by generating and composing programs. The results indicate improved performance over prior approaches and could be interesting to the RL community.

**Weaknesses:**

I initially lean toward rejection due to two primary concerns:

(1) *Unclear presentation and missing details.*
- Several key pieces of information are not clearly described in the main text of the paper, which makes it difficult to understand the exact proposed approach and to reproduce the results:
- How are the programs over which the embedding space is learned generated? Are they simply randomly sampled from the Karel DSL? These programs form such a crucial building block for the method that it's worth discussing how they're generated here, and how they might be generated in other domains.
- How are the rewards defined in the CEM optimization in the retrieval stage? It seems like one would need many different reward functions to ensure good diversity and coverage in the selected modes. Is it task-specific? This crucial information does not appear to be clearly explained anywhere.
- How is the transition function f learned? Is this learned via standard RL, where choosing the modes is the action?
- How do state machine policies differ from other notions of hierarchical reinforcement learning? It seems like the modes can simply be interpreted as "skills" or formally options in a SMDP framework.
- Details about the state are missing from the main body of the work, which makes it difficult to understand the task. From what I gather from the Appendix, a N x M array is used as the state for a N x M grid world for the deep RL baseline -- is this the same state that the proposed approach uses for selecting which mode to switch to?
- The deep RL baseline seems to be surprisingly weak, given that the task is just grid world pick and place. Is there a reason behind this? Is it a result of the state representation? Does e.g., using a different state representation of just the agent's (x, y) coordinates work better?

(2) *Concerns about the generality of the proposed approach.*
- While the details for how the entire space of programs is generated is missing, it seems difficult to imagine being able to do this well for arbitrary domains. In general, writing programs that can easily compose is challenging, particularly when they have inputs and outputs, whereas this domain is particularly simple and the programs do not need to take arguments or return values. Consequently, I'm concerned that the proposed approach may not be able to be applied to other less toy domains. Discussion about this would be very helpful.
- It also seems generally difficult to span a set of useful behaviors needed in the programs. How can this be achieved without ending up with an intractable space of programs to search over?

**Questions:**

Please see the many questions in the previous section.

---

> ### Author Response · Authors · 2023-11-22
> **Response to Reviewer NAAH (1/2)**
>
> We thank the reviewer for the thorough and constructive comments. Please find the response to your questions below.
>
> ### Responses to Questions
>
> > Unclear presentation and missing details
>
> We sincerely thank the reviewer for pointing these out. We have thoroughly revised the paper to include the missing details and make the existing information easier to find.
>
> > How are the programs over which the embedding space is learned generated? Are they simply randomly sampled from the Karel DSL? These programs form such a crucial building block for the method that it's worth discussing how they're generated here, and how they might be generated in other domains.
>
> As described in Section 4.1, we follow the training procedure proposed by LEAPS (Trivedi et al., 2021), which generates 50000 programs that are randomly sampled according to the Karel DSL to train the encoder-decoder model. Further details about the program dataset generation and the training of the encoder-decoder model can be found in Section F.1 of the revised paper.
>
> To generate programs in another domain, we can simply randomly sample programs according to the DSL in that domain "for free" and then learn a program embedding space.
>
> > How are the rewards defined in the CEM optimization in the retrieval stage? It seems like one would need many different reward functions to ensure good diversity and coverage in the selected modes. Is it task-specific? This crucial information does not appear to be clearly explained anywhere.
>
> The reward used by POMP for CEM optimization, as shown in Equation 1 and Figure 3, is defined as:
>
> $R\_{\rho} = \sum\_{i=1}^{|\Psi\_{\text{before}}|} \gamma ^ {i-1} \sum\_{t=0}^{T^i} \gamma^t \mathbb{E}\_{(s\_t,a\_t) \sim \text{EXEC} (\Psi\_{\text{before}}[i])}[r\_{t}] + \gamma^{|\Psi\_{\text{before}}|}\sum\_{t=0}^{T} \gamma^t \mathbb{E}\_{(s\_t,a\_t) \sim \text{EXEC} (\rho\_z)}[r\_{t}]  + \gamma^{|\Psi\_{\text{before}}|+1}\sum\_{i=1}^{|\Psi\_{\text{after}}|} \gamma^{i-1} \sum\_{t=0}^{T^i} \gamma^t \mathbb{E}\_{(s\_t,a\_t) \sim \text{EXEC} (\Psi\_{\text{after}}[i])}[r\_{t}].$
>
> Note that this reward function is not task-specific, and we use this reward to retrieve the set of mode programs for POMP across all tasks discussed in Section 5.1.
>
> > How is the transition function f learned? Is this learned via standard RL, where choosing the modes is the action?
>
> As stated in Section 4.3 and Section F, the learning process of transition function $f$ is formulated as a standard RL problem. A state includes the current Karel grid and the previously executed mode represented by a one-hot vector, while an action chooses the next mode program, as correctly understood by the reviewer. We use the PPO algorithm to optimize the transition function $f$.
>
> > How do state machine policies differ from other notions of hierarchical reinforcement learning? It seems like the modes can simply be interpreted as "skills" or formally options in a SMDP framework.
>
> We fully agree with the reviewer that our proposed Program Machine Policy shares the same spirit and some ideas with HRL frameworks and SMDP if we view the transition function as a “high-level” policy and the set of mode programs as “low-level” policies or skills.
>
> While most HRL frameworks either pre-define and pre-learned low-level policies, or jointly learn the high-level and low-level policies from scratch, our proposed framework first retrieves a set of effective, diverse, and compatible modes, and then learns the mode transition function.
>
> We have revised the paper and included a discussion on HRL (Section A). We would appreciate it if the reviewer could suggest more related HRL and SMDP papers if our discussion misses anything.
>
> > Details about the state are missing from the main body of the work, which makes it difficult to understand the task. From what I gather from the Appendix, a N x M array is used as the state for a N x M grid world for the deep RL baseline -- is this the same state that the proposed approach uses for selecting which mode to switch to?
>
> We thank the reviewer for bringing up this issue. Yes, as described in Section 4.3 and Section F of the revised paper, both the DRL baseline and the mode transition function of POMP use a CNN to encode an input state of an N x M array, describing the current Karel grid, as shown in Figure 5 and Section H of the revised paper.

---

> > ### Author Response · Authors · 2023-11-22
> > **Response to Reviewer NAAH (2/2)**
> >
> > > The deep RL baseline seems to be surprisingly weak, given that the task is just grid world pick and place. Is there a reason behind this? Is it a result of the state representation? Does e.g., using a different state representation of just the agent's (x, y) coordinates work better?
> >
> > We thank the reviewer for mentioning this observation. DRL struggles at the tasks in the Karel-Long problem set that require long-horizon exploration with sparse rewards (i.e., Seesaw and Up-N-Down), as described in Section I. On the other hand, DRL can achieve satisfactory performance on the tasks which provide dense rewards (i.e., Farmer, Inf-DoorKey, and Inf-Harvester), even when the episode horizon is long.
> >
> > > While the details for how the entire space of programs is generated is missing, it seems difficult to imagine being able to do this well for arbitrary domains. In general, writing programs that can easily compose is challenging, particularly when they have inputs and outputs, whereas this domain is particularly simple and the programs do not need to take arguments or return values. Consequently, I'm concerned that the proposed approach may not be able to be applied to other less toy domains. Discussion about this would be very helpful.
> >
> > We thank the reviewer for raising this concern. We certainly agree with the reviewer that the proposed framework, and most programmatic reinforcement learning methods, are not generally applicable to arbitrary domains.
> >
> > Our program policies are designed to describe high-level task-solving procedures or decision-making logics of an agent. Therefore, our principle of designing domain-specific languages (DSLs) considers a general setting where an agent can perceive and interact with the environment to fulfill some tasks. DSLs consist of control flows, perceptions, and actions. While control flows are domain-independent, perceptions and actions can be designed based on the domain of interest, which would require specific expertise and domain knowledge.
> >
> > Such DSLs are proposed and utilized in various domains, including ViZDoom (Kempka et al., 2016), 2D Minecraft (Sun et al., 2020), and gym-minigrid (Chevalier-Boisvert et al., 2023). Recent works (Liang et al., 2023, Wang et al., 2023) also explore describing agents’ behaviors using programs with functions taking arguments. We have revised the paper to include this discussion (Section J).
> >
> > > It also seems generally difficult to span a set of useful behaviors needed in the programs. How can this be achieved without ending up with an intractable space of programs to search over?
> >
> > Following LEAPS (Trivedi et al., 2021) and HPRL (Liu et al., 2023), we construct a program embedding space that continuously parameterizes diverse programs by randomly generating a program dataset. We limit the program length when generating the dataset, which subsequently confines the possible program space not to be unlimitedly large or intractable, and therefore the dataset programs describe shorter-horizon behaviors.
> >
> > As pointed out by the reviewers, the majority of the generated programs do not capture meaningful behaviors. Therefore, devising efficient search methods that can quickly identify effective programs is critical, which is the main focus of LEAPS and HRPL. The search method proposed in our work, on the other hand, not only aims to obtain effective programs but is also designed to retrieve a set of effective, diverse, and compatible programs. For example, Figure 7 illustrates how the proposed CEM+diversity can avoid searching over paths similar to those previously explored, which drastically increases the searching efficiency.
> >
> > ### References
> >
> > - Trivedi et al. “Learning to Synthesize Programs as Interpretable and Generalizable Policies” in NeurIPS 2021
> > - Kempka et al. "Vizdoom: A doom-based ai research platform for visual reinforcement learning" in IEEE Symposium on Computational Intelligence and Games 2016
> > - Sun et al. ‘Program Guided Agent” in ICLR 2020
> > - Chevalier-Boisvert, et al. “Minigrid & Miniworld: Modular & Customizable Reinforcement Learning Environments for Goal-Oriented Tasks’” arxiv 2023
> > - Liang et al. “Code as Policies: Language Model Programs for Embodied Control” in ICRA 2023
> > - Wang et al. “Demo2Code: From Summarizing Demonstrations to Synthesizing Code via Extended Chain-of-Thought” arxiv 2023
> > - Liu et al. “Hierarchical Programmatic Reinforcement Learning via Learning to Compose Programs” in ICML 2023
> >
> > ### Conclusion
> >
> > We are incredibly grateful to the reviewer for the detailed and constructive review. We believe our responses address the concerns raised by the reviewer. Please kindly let us know if there are any further concerns or missing experimental results that potentially prevent you from accepting this submission. We would be more than happy to address them if time allows. Thank you very much for all your detailed feedback and the time you put into helping us to improve our submission.

---

### Author Response · Authors · 2023-11-22
**Paper Revision**

We thank all the reviewers for their thorough and constructive comments. We have revised the paper to include these discussions. The major changes are summarized as follows.

1. **[Related works]** We thank the reviewers for suggesting various related works. In response, we have updated our paper to include discussions on them.
- Learning Finite State Representations Recurrent Policy Networks (Reviewer HDvE)
- Show Me the Way! Bilevel Search for Synthesizing Programmatic Strategies (Reviewer HDvE)
- Hierarchical reinforcement learning with the MAXQ value function decomposition (Reviewer dxYr)
- Integrating symbolic planning and skill policies (Reviewer bA3y)
    - LEAGUE: Guided Skill Learning and Abstraction for Long-Horizon Manipulation
    - Leveraging approximate symbolic models for reinforcement learning via skill diversity
    - PEORL: Integrating Symbolic Planning and Hierarchical Reinforcement Learning for Robust Decision-Making

2. **[Relation to Hierarchical Reinforcement Learning]** (Reviewer NAAH and Reviewer dxYr) We have included an extended related work section that provides better contextualization and discussion about the proposed framework for hierarchical reinforcement learning (HRL) and option frameworks.

3. **[New Baselines]** (Reviewer HDvE) As suggested by Reviewer HDvE, we have added two additional baselines in Table 2.
- Programmatic state machine policy (PSMP), whose modes are primitive actions of the agent instead of mode programs retrieved by our method, resembling the idea of Inala et al., 2020
- Moore machine network (MMN; Koul et al., 2019)

4. **[Experimental Results of Karel and Karel-Hard Problem Sets]** (Reviewer HDvE and Reviewer dxYr) We have included the results of our proposed POMP on the Karel and Karel-Hard tasks in Table 3, which demonstrate the effectiveness of POMP. In short, POMP outperforms LEAPS and HPRL on most Karel and Karel Hard tasks by large margins.

5. **[Experimental Result Discussions]** (Reviewer HDvE and Reviewer dxYr) We have included additional discussions on the experimental results of our proposed framework, DRL, HPRL, PSMP, and MMN.

6. **[Improved Interpretability]** (Reviewer HDvE) Following the suggestion of Reviewer HDvE, we have extracted Moore machines from our learned transition function by adopting the approach proposed in Koul et al., 2019. We illustrated the extracted state machines in Section E of the revised paper.

7. **[Clarification]**
- Learning a program embedding space in Section F (Reviewer NAAH)
- How to compute the proposed compatibility bonus in Section 4.2.3 (Reviewer bA3y and Reviewer dxYr)
- Program sample efficiency in Section 5.6 and Section C (Reviewer HDvE)

---

### Meta-Review · Area_Chair_B54j · 2023-12-09

**Metareview:**

This paper studies the problem of programmatic reinforcement learning, which aims to learn interpretable policies in the form of programs. Their approach is to first synthesize programs representing short term behaviors relevant to solving the long-horizon task, while training to ensure that these programs are "compatible" with one another in some way. Then, their approach learns a high-level state machine policy that "stitches" these programs together to solve the overall task.

Overall, the reviewers agree that the approach proposed by the authors is an interesting one, and hierarchical policies of this form are an important step towards scaling programmatic reinforcement learning towards more challenging problems. However, there were a range of concerns about potential limitations of the approach, including arguments about interpretability, the potential for execution incompatibilities, etc. The most salient concern was about the generality of the proposed approach; the reviewers largely agreed that the domains used in the experiments were too toy in nature, and evaluating on more challenging domains is critical to demonstrating that the proposed approach has the potential to generalize.

**Justification For Why Not Higher Score:**

While each reviewer had different concerns, they all agreed that the evaluation was a major weakness, and that the approach needs to be evaluated on more challenging domains to demonstrate generalizability.

**Justification For Why Not Lower Score:**

N/A

---

### Decision · Program_Chairs · 2024-01-16

Reject